# Structural analysis of the dynamic ribosome-translocon complex

Aaron JO Lewis[1], Frank Zhong[2], Robert J Keenan[3], Ramanujan S Hegde[1]*

[1]MRC Laboratory of Molecular Biology, Cambridge, United Kingdom; [2]Department of Molecular Genetics and Cell Biology, The University of Chicago, Chicago, United States; [3]Department of Biochemistry and Molecular Biology, The University of Chicago, Chicago, United States

**Abstract** The protein translocon at the endoplasmic reticulum comprises the Sec61 translocation channel and numerous accessory factors that collectively facilitate the biogenesis of secretory and membrane proteins. Here, we leveraged recent advances in cryo-electron microscopy (cryo-EM) and structure prediction to derive insights into several novel configurations of the ribosome-translocon complex. We show how a transmembrane domain (TMD) in a looped configuration passes through the Sec61 lateral gate during membrane insertion; how a nascent chain can bind and constrain the conformation of ribosomal protein uL22; and how the translocon-associated protein (TRAP) complex can adjust its position during different stages of protein biogenesis. Most unexpectedly, we find that a large proportion of translocon complexes contains RAMP4 intercalated into Sec61's lateral gate, widening Sec61's central pore and contributing to its hydrophilic interior. These structures lead to mechanistic hypotheses for translocon function and highlight a remarkably plastic machinery whose conformations and composition adjust dynamically to its diverse range of substrates.

## eLife assessment

This **landmark** work by Lewis et al. represents the most significant breakthrough in membrane and secretory biogenesis in recent years. Their work reveals with outstanding clarity how nascent transmembrane segments can pass through the gate of Sec61 into the ER membrane through the coordinated motions of a conformationally and compositionally dynamic machine. Among many other insights, the authors discovered how a new factor, RAMP4, contributes to the formation and function of the lateral gate for certain substrates. The technical quality of the work is **exceptional**, setting the bar appropriately high.

*For correspondence:
rhegde@mrc-lmb.cam.ac.uk

## Introduction

Most eukaryotic secretory and membrane proteins are translocated co-translationally across the endoplasmic reticulum (ER) membrane at a ribosome-translocon complex (RTC). The central component of this translocon is the Sec61 complex, a heterotrimer containing a channel-forming α subunit and peripheral β and γ subunits (*Van den Berg et al., 2004*; *Rapoport et al., 2017*). In prokaryotes, the homologous SecYEG complex mediates translocation across the plasma membrane. In all organisms, this central channel associates dynamically with a variety of partners and accessory factors (*Gemmer and Förster, 2020*; *Shao, 2023*; *Hegde and Keenan, 2022b*; *Smalinskaitè and Hegde, 2023*). The structure and function of some accessory factors, such as the oligosaccharyl transferase complex (OST), are well established (*Braunger et al., 2018*), whereas many others are poorly understood (*Gemmer and Förster, 2020*; *Shao, 2023*; *Hegde and Keenan, 2022b*; *Smalinskaitè and Hegde, 2023*). Although these various translocon components have long been speculated to engage

 

in a context- and substrate-dependent manner (*Johnson and van Waes, 1999*; *Hegde and Kang, 2008*), the rules governing their coordination are largely unclear and only now beginning to emerge (*Smalinskaitė et al., 2022*; *Sundaram et al., 2022*).

Sec61α is a pseudosymmetric channel that can open axially across the membrane and is laterally gated towards the lipid bilayer (*Van den Berg et al., 2004*; *Rapoport et al., 2017*). In its closed state, the axial channel's pore is constricted, blocked by a short helix known as the plug, and the lateral gate is closed. Transport of hydrophilic polypeptide segments through Sec61 can be initiated by a flanking hydrophobic α-helix. These hydrophobic helices act as signal sequences that bind to Sec61's lateral gate (*Voorhees and Hegde, 2016a*; *Li et al., 2016*). This binding is thought to widen the central pore, destabilise the plug, and thread one of the signal's hydrophilic flanking regions into the channel to initiate translocation. A set of hydrophobic residues, known as the pore ring, lines the narrowest part of the channel and forms a gasket-like seal that maintains the permeability barrier during translocation (*Dalal and Duong, 2009*; *Park and Rapoport, 2011*; *Ma et al., 2019*).

This model of channel opening is derived from structures of a cleavable signal peptide (SP) bound to the mammalian Sec61 complex (*Voorhees and Hegde, 2016a*) or bacterial SecYEG (*Li et al., 2016*; *Ma et al., 2019*). Most membrane proteins lack an N-terminal SP and instead initiate translocation using their first TMD, often termed a signal anchor (SA). Although it has long been thought an SA initiates flanking domain translocation the same way as an SP (*High et al., 1993a*; *Görlich and Rapoport, 1993*; *Mothes et al., 1998*; *Heinrich et al., 2000*), direct evidence for this idea is sparse and somewhat contradictory. Furthermore, the route an SP or SA takes to the lateral gate remains speculative. One model posits that they all access the lateral gate from the channel interior, displacing the plug en route. Alternatively, a hydrophobic helix could slide along the lipid-facing side of the lateral gate (*Cymer et al., 2015*), or use a member of the Oxa1 family for insertion (*Anghel et al., 2017*; *Chitwood et al., 2018*; *Wu and Hegde, 2023*). Structures of TMD insertion intermediates would help resolve these and other crucial mechanistic issues in membrane protein biogenesis.

Beyond the Sec61 channel, a number of accessory factors co-translationally modify the polypeptide, facilitate membrane protein biogenesis, or facilitate secretion through Sec61. The two major modification factors are OST (*Braunger et al., 2018*) and the signal peptidase complex (SPC) (*Liaci et al., 2021*), which mediate co-translational N-linked glycosylation and SP cleavage, respectively. Based on recent structural and functional studies (*Smalinskaitė et al., 2022*; *Sundaram et al., 2022*; *Chitwood et al., 2018*; *Wu and Hegde, 2023*; *McGilvray et al., 2020*; *Chitwood and Hegde, 2020*; *Shurtleff et al., 2018*; *Gemmer et al., 2023b*), and consistent with genetic co-dependency analysis across more than a thousand cancer cell lines (*Meyers et al., 2017*; *Wainberg et al., 2021*; *Dempster et al., 2019*), four protein complexes are now assigned to co-translational membrane protein biogenesis: EMC, PAT, GEL, and BOS. The functions of these complexes and the dynamics of their association with nascent substrates at the ribosome are not well understood.

EMC is a 9-protein complex conserved across eukaryotes and implicated in diverse aspects of membrane protein biogenesis (*Hegde, 2022a*). At least one main function of EMC is to facilitate insertion of TMDs close to the N- or C-terminus. This includes post-translational insertion of tail-anchored proteins (*Guna et al., 2018*), post-translational insertion of the final TMD of some multi-pass membrane proteins (*Wu et al., 2024*), and co-translational insertion of SAs in the $N_{exo}$ topology (N-terminus facing the exoplasmic side of the membrane) (*Chitwood et al., 2018*; *O'Keefe et al., 2021*). This insertase function is thought to involve its core EMC3 subunit (*Wu and Hegde, 2023*; *Wu et al., 2024*; *Pleiner et al., 2020*; *Miller-Vedam et al., 2020*; *Bai et al., 2020*; *Güngör et al., 2022*), which is a member of the Oxa1 family of insertases (*Anghel et al., 2017*). Although EMC has been implicated in co-translational membrane protein insertion, it has thus far not been observed at RTCs in cryo-electron tomography studies (*Gemmer et al., 2023b*; *Gemmer et al., 2023a*).

The PAT, GEL, and BOS complexes are recruited to the ribosome-Sec61 complex at a later stage of multipass biogenesis to form the multipass translocon (MPT) (*Smalinskaitė et al., 2022*; *Sundaram et al., 2022*; *McGilvray et al., 2020*; *Chitwood and Hegde, 2020*). The MPT is thought to facilitate membrane protein biogenesis by a combination of TMD insertion, chaperoning, and shielding (*Hegde and Keenan, 2022b*; *Smalinskaitė and Hegde, 2023*). The MPT-mediated insertion reaction is thought to operate on pairs of TMDs connected by a short translocated loop. This insertion might be facilitated by the GEL complex, whose TMCO1 subunit is another member of the Oxa1 family (*Anghel et al., 2017*), and does not seem to rely on the lateral gate of the Sec61 complex (*Smalinskaitė*

*et al., 2022*). TMDs with partially polar character are thought to be chaperoned by the PAT complex (*Chitwood and Hegde, 2020*) via an amphiphilic surface on its Asterix subunit (*Smalinskaitė et al., 2022*). The function of the BOS complex is unclear, but its contacts with Sec61 and the ribosome may facilitate MPT assembly or stability (*Smalinskaitė et al., 2022*; *McGilvray et al., 2020*). The PAT, GEL, and BOS complexes together form a horseshoe-shaped lipid-filled cavity (*Smalinskaitė et al., 2022*; *McGilvray et al., 2020*; *Gemmer et al., 2023b*; *Gemmer et al., 2023a*) speculated to be the site of multipass membrane protein folding (*Hegde and Keenan, 2022b*; *Smalinskaitė and Hegde, 2023*).

Three factors are linked to co-translational Sec61-mediated secretion: TRAM (*Görlich et al., 1992*; *Voigt et al., 1996*; *Hegde et al., 1998*), TRAP (*Fons et al., 2003*), and RAMP4 (also called SERP1, known as Ysy6 in yeast) (*Görlich and Rapoport, 1993*; *Schröder et al., 1999*; *Yamaguchi et al., 1999*; *Sakaguchi et al., 1991*). Both TRAP and RAMP4 are tightly associated near-stoichiometrically with ER-localised ribosome-Sec61 complexes from pancreas (*Görlich and Rapoport, 1993*), an exceptionally secretory tissue. TRAM, although not tightly associated with ribosomes, can be crosslinked to SPs and TMDs at a point when they are at Sec61's lateral gate (*Görlich et al., 1992*; *High et al., 1993b*; *Jungnickel and Rapoport, 1995*). Depletion of TRAM or TRAP reduces the capability of some SPs to successfully initiate translocation through Sec61 in vitro, placing their function at an early gating step (*Görlich et al., 1992*; *Voigt et al., 1996*; *Fons et al., 2003*). The role of RAMP4 is largely unclear, but it can be crosslinked to nascent chains translocating through Sec61 (*Schröder et al., 1999*), shows secretion defects in knockout cells and mice (*Hori et al., 2006*), and it is induced by ER stress (*Yamaguchi et al., 1999*). The mechanisms by which any of these factors impacts secretion are unclear, but would be aided by structural information on how they engage the ribosome-Sec61 translocon.

In this study, we have taken advantage of the heterogeneity of most stalled protein biogenesis intermediates assembled in vitro. Whereas substrates at certain key steps might have uniform and stable interactions with the biogenesis machinery (*Voorhees and Hegde, 2016a*), others probably sample multiple states dynamically (*Sundaram et al., 2022*). For example, both OST and MPT factors dynamically and heterogeneously associate at their overlapping sites behind Sec61. TMDs sample multiple environments such as the lateral gate, intramembrane chaperones, and the surrounding lipid. Hence, a substrate stalled at a single point during elongation can form multiple RTCs. Improvements in cryo-electron microscopy (cryo-EM) imaging and particle classification now allow sample heterogeneity to be resolved into multiple discrete density maps (*Scheres, 2016*; *von Loeffelholz et al., 2017*). When combined with rapid advances in structure prediction (*Jumper et al., 2021*; *Evans et al., 2021*; *Mirdita et al., 2022*), these maps can be fitted with reliable models. Using this approach, we present structures of a TMD insertion intermediate at Sec61's lateral gate, RAMP4 and TRAP within RTCs, and a new configuration of ribosomal protein uL22 that forms contacts with Sec61 and the nascent substrate.

## Results and discussion

We performed a thorough single-particle cryo-EM analysis of a previous dataset of RTCs engaged in biogenesis of the multipass membrane protein rhodopsin (Rho) (*Smalinskaitė et al., 2022*). The construct is termed Rho[ext] because it contains the first two TMDs of Rho and is extended at its N-terminus by fusion to an SP and an epitope tag (*Figure 1A*). The Rho[ext] intermediate in this sample has elongated to the point where the SP has been removed after directing translocation of the N-terminal domain across the membrane, RhoTM1 has inserted into the membrane, and RhoTM2 has emerged from the ribosome (*Smalinskaitė et al., 2022*). At this critical chain length, the nascent chain is poised at a point where it can potentially form several different RTCs (*Figure 1B–E*; *Figure 1—figure supplement 1*), only one of which was analysed previously (*Smalinskaitė et al., 2022*).

Crosslinking and co-association experiments (*Smalinskaitė et al., 2022*; *Sundaram et al., 2022*; *Chitwood and Hegde, 2020*) show that the nascent chain is just long enough for RhoTM1 to begin recruiting the PAT complex and initiate assembly of the MPT. Analysis of this subset of RTCs previously showed how the PAT complex latches Sec61 shut and redirects RhoTM2 towards the just-assembling MPT (*Figure 1E*; *Smalinskaitė et al., 2022*). Because PAT complex recruitment and MPT assembly are just beginning, the dataset also contains multiple PAT-free complexes. We now analyse these particles via extensive classification (*Figure 1—figure supplement 1*) and find that without PAT, Sec61 is either closed, opened by RhoTM2, or opened by an unexpected factor, RAMP4 (*Figure 1B–D*). The closed structure is similar to the previously reported PAT-bound structure, so we focus on RhoTM2- and

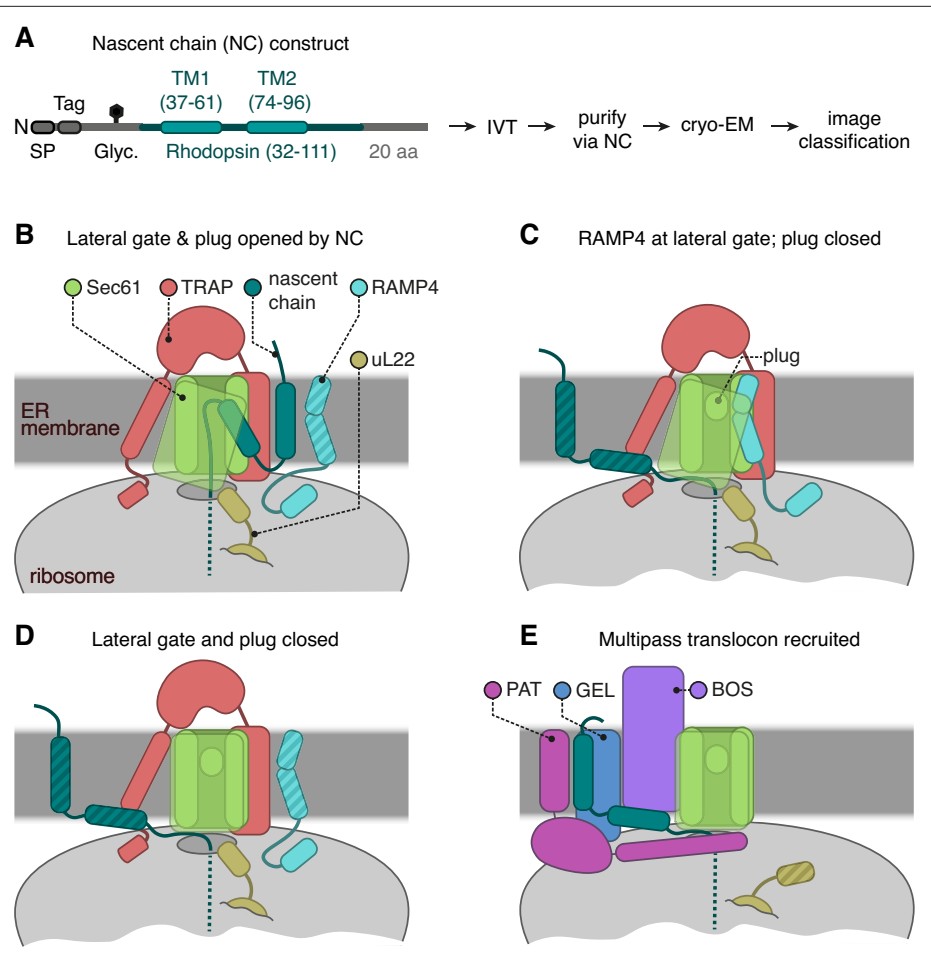

**Figure 1.** Cryo-electron microscopy (cryo-EM) analysis of ribosome-translocon complexes (RTCs). (**A**) Diagram of the Rho^ext construct (not to scale) containing the first two transmembrane domains (TMDs) and flanking regions of bovine rhodopsin (amino acids 32–111, Uniprot ID P02699). The rhodopsin region is preceded by a signal peptide (SP), an epitope tag, and a polypeptide of 52 amino acids containing a site for N-linked glycosylation (Glyc.) to monitor translocation. The experimental strategy used in earlier work (*Smalinskaitė et al., 2022*) to generate and analyse structurally the intermediates of Rho^ext is indicated. (**B–E**) The four major classes of RTCs observed after image classification (*Figure 1—figure supplement 1*). The multipass translocon class in panel (**E**) was reported previously (*Smalinskaitė et al., 2022*). The current work presents RTCs represented in panels (**B and C**). The closed translocon, which essentially combines the closed Sec61 state seen in panel (**E**) with the subunit composition seen in panel (**B**), is not discussed separately here. The hatched RTC elements (e.g. the RAMP4 membrane domain in panel **B**) indicate regions that are not visible in the cryo-EM maps but are inferred to be present from other data.

The online version of this article includes the following figure supplement(s) for figure 1:

**Figure supplement 1.** Cryo-electron microscopy (cryo-EM) image processing.

**Figure supplement 2.** Local and global map resolutions.

RAMP4-bound structures. We then compare various available structures, together with structure predictions and modelling, to reveal new insights into the structural dynamic of RTCs.

## The structure of Sec61 inserting a membrane protein

In the absence of the PAT complex, the Sec61 complex is free to open. In the intermediate being analysed, RhoTM2, whose eventual topology in the final protein would be $N_{cyt}$ (i.e. the N-terminal flanking domain facing the cytosol), has just fully emerged from the ribosome with a 33 amino acid downstream tether to the P site tRNA. This is long enough for the first half of RhoTM2 to begin engaging the Sec61 complex, and a subset of RTCs displayed this event (*Figure 2A and B*). We find that the

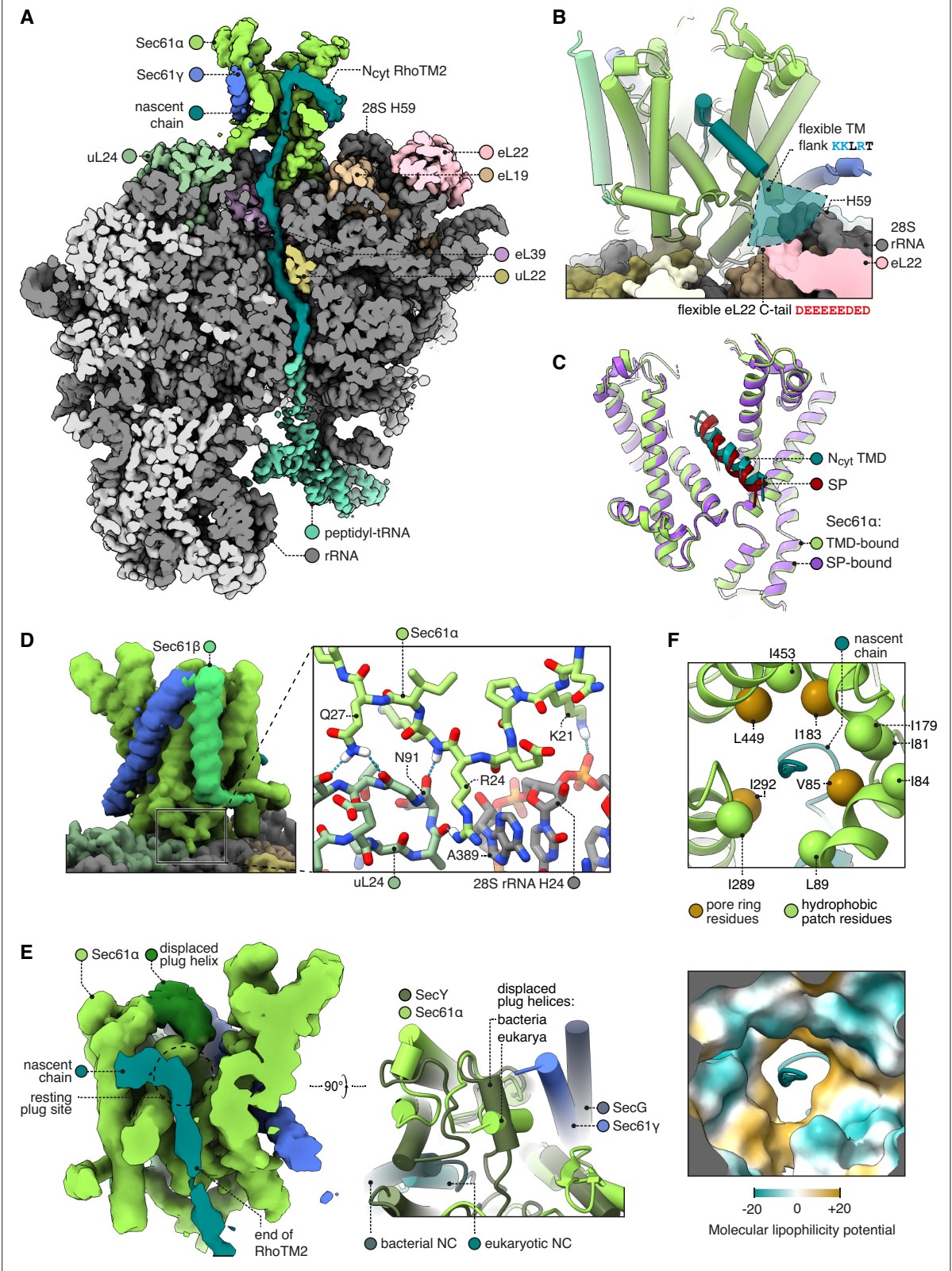

**Figure 2.** Structure of Sec61 bound to an $N_{cyt}$ transmembrane domain (TMD). (**A**) Overview of the $N_{cyt}$ RhoTM2-bound Sec61 complex. For display, the density map was filtered using DeepEMhancer and clipped in plane with the ribosomal exit tunnel. (**B**) Close-up view of the Sec61-RhoTM2 complex. A semitransparent circular sector stands in for the flexible TM-flanking loop, which contains basic residues likely to interact with two nearby polyacidic parts of the ribosome: 28S rRNA helix 59 and the C-tail of eL22. (**C**) $N_{cyt}$ TMD-bound Sec61 is structurally similar to signal peptide (SP)-

*Figure 2 continued on next page*

*Figure 2 continued*

bound Sec61. The SP is preprolactin (PDB 3JC2 rebuilt into EMD-3245 using restraints from AF2). (**D**) The Sec61 N-half binds the ribosome. (**E**) When displaced by a translocating nascent chain, the plug helix relocates to a lumenal site near Sec61γ in eukaryotes and SecG in bacteria, but its orientation is not conserved. For display, the density map was filtered using DeepEMhancer and clipped in plane with the channel pore. (**F**) Structure of the pseudosymmetric pore ring residues V85, I183, I292, and L449.

The online version of this article includes the following figure supplement(s) for figure 2:

**Figure supplement 1.** Comparison of the pore ring across different models of Sec61.

N-terminal half of RhoTM2 binds to Sec61 very similarly to the previously characterised hydrophobic helix (h-region) of an SP (*Figure 2C*; *Voorhees and Hegde, 2016a*; *Li et al., 2016*). Thus, both kinds of $N_{cyt}$ substrate, SPs and TMDs, can bind Sec61 in the same way, despite TMDs having much longer hydrophobic helices than SPs do. The extra length of this TMD is passing through the channel pore, as will be discussed below.

Bound in this position, the N-terminal end of the hydrophobic region of the SP or TMD is only 11 Å (~3 aa) from the tip of 28S rRNA helix 59 (H59) and the adjacent polyacidic tail of eL22 (*Figure 2B*). The backbone phosphates and the exposed aromatic rings of U2707-8 in H59, and the acidic residues of eL22, are potential binding sites for basic side chains. Importantly, basic resides are sharply enriched ~3–5 aa from the hydrophobic helix (*Baker et al., 2017*). Thus, H59 and eL22 are well positioned to engage cationic residues flanking a signal and retain them in the cytosol, a phenomenon called the positive-inside rule (*Nilsson et al., 2005*). Notably, the next nearest rRNA, H47, is too far to serve this role (32 Å, ~9 aa). Earlier lower resolution maps of a bacterial RTC similarly noted the proximity of H59 to the membrane and SecY's lateral gate (*Frauenfeld et al., 2011*). Given that H59 is also near the N-terminus of an SP bound to bacterial and mammalian signal recognition particle (SRP) (*Jomaa et al., 2016*; *Kobayashi et al., 2018*), H59 may contribute to the positive-inside rule throughout both targeting by SRP and insertion by Sec61.

To accommodate the TMD, the N-half of Sec61 has rotated out of the membrane plane towards the ribosome (*Figure 2D*). It is striking that the open conformation induced by the TMD is indistinguishable from that induced by an SP (Cα RMSD 0.691 Å; *Figure 2C*), despite their dissimilar sequences and despite Sec61's continuous flexibility (*Mercier et al., 2021*; *Itskanov et al., 2021*). As an explanation for this bistable behaviour, we observe that the open conformation is stabilised by contacts between the N-half of Sec61α and the ribosome (*Figure 2D*). The N-half and ribosome have been thought to be isolated from one another, but we find that Sec61 residues 21–27 contact the 28S rRNA helix 24 and uL24, including a particularly well-resolved cation-π interaction between Sec61α R24 and 28S A389. Bacterial SecY lacks this entire loop, perhaps because bacterial secretion is driven by SecA, which competes with ribosomes (*Rapoport et al., 2017*; *Wu et al., 2012*; *Zimmer et al., 2008*). Archaeal SecY, however, does conserve this loop's structure (Cα RMSD 0.67 Å with *Methanocaldococcus jannaschii*) and consensus sequence (euk. KPERKIQ vs arc. KPERKVSL, with the cation-π arginine underlined). Stabilising interactions with this widely conserved motif may help Sec61 respond to its diverse substrates with a consistent open state.

Whereas the N-terminal half of RhoTM2 is helical, its C-terminal half loops back through the Sec61 pore in an unfolded conformation similar to the segment of polypeptide downstream of an SP during secretion (*Voorhees and Hegde, 2016a*; *Ma et al., 2019*). This is accommodated by the plug helix moving towards the lumenal tip of Sec61γ, effectively becoming part of the channel's lumenal funnel (*Figure 2E*), as previously seen with the bacterial plug (*Li et al., 2016*; *Ma et al., 2019*). Our observation that different segments of a TMD can simultaneously occupy the lateral gate and central channel is consistent with the through-pore model of insertion, rather than the sliding model (*Cymer et al., 2015*), at least for this substrate. The through-pore translocation may be favoured because the ribosome exit tunnel's mouth holds an emerging TMD closer to the Sec61 pore than to the lateral gate. Moreover, the free energy cost of passing a hydrophobic TMD through the hydrophilic pore could be mitigated by the hydrophobic pore ring and partly hydrophilic TMD.

Prior structures of ribosome-bound, open Sec61 disagreed on the structure of the channel pore due to differences in how the rotations of core helices were modelled (*Figure 2—figure supplement 1*; *Braunger et al., 2018*; *Voorhees and Hegde, 2016a*). With ~3.7 Å resolution (*Figure 1—figure supplement 2A*) and 1.8–3.5 Å predicted aligned error (PAE) restraints from the AlphaFold2 (AF2) prediction of Sec61, we find that the pore is ringed by four aliphatic residues: V85, I183, I292, and

L449. Nearby residues F42, I81, I84, L89, I179, I187, I289, and I453 extend hydrophobic patches outwards from the pore ring (*Figure 2F*). The four pore ring residues identified here are homologous to the four pore ring residues of bacterial SecY (*Dalal and Duong, 2009*; *Ma et al., 2019*; *Lewis and Hegde, 2021*), and adopt similar positions as in the open SecY structure (Cα RMSD 1.1 Å) (*Ma et al., 2019*). These four residues are pseudosymmetric and likely inherited from SecY's dimeric ancestor (*Lewis and Hegde, 2021*). Thus, this structure shows that the likely ancestral pore architecture is universally conserved across the SecY family.

The structure of RhoTM2 trapped during insertion via the central channel of Sec61 represents a paradigm for insertion of $N_{cyt}$ SAs and the subset of $N_{cyt}$ TMDs in multipass membrane proteins followed by a long (>100 aa) translocated loop. Both types of substrates are potently inhibited by Sec61 inhibitors (*Smalinskaitè et al., 2022*; *O'Keefe et al., 2021*; *Zong et al., 2019*; *McKenna et al., 2017*; *Morel et al., 2018*; *Maifeld et al., 2011*; *Paatero et al., 2016*; *Tranter et al., 2020*) whose binding site at the lateral gate is mutually exclusive with the position of the TMD observed in our structure (*Itskanov et al., 2023*; *Rehan et al., 2023*). In the case of $N_{cyt}$ TMDs followed by a short (<50 aa) translocated loop and another TMD, insertion is thought to occur as a TMD pair via an Oxa1 family member such as the GEL complex within the MPT (*Hegde and Keenan, 2022b*; *Smalinskaitè and Hegde, 2023*; *Smalinskaitè et al., 2022*; *Sundaram et al., 2022*). Given that Rho biogenesis is not sensitive to Sec61 lateral gate inhibitors, RhoTM2 seems to normally be inserted together with RhoTM3 via the GEL complex (*Smalinskaitè et al., 2022*). Although the structure seen here probably represents an alternative route captured due to ribosome stalling, it nonetheless proved to be an illuminating paradigm.

## RAMP4 occupies the Sec61 gate

Alongside a RhoTM2-bound map, image classification yielded a map in which Sec61 is bound to a different and unexpected density. This density consists of a kinked TMD bound to the Sec61 lateral gate and a ribosome-binding domain (RBD; *Figure 3A*). The resolution of the RBD density was sufficient for de novo modelling, identifying the sequence as RAMP4. The location, structure, and function of RAMP4 has long been unclear.

RAMP4's RBD is hook-shaped (*Figure 3B*). Its N-terminus is comprised of an α-helix (residues 5–15) flanked by $3_{10}$-helices (3–7,13–20). Subsequent residues then loop back along the helix. The hook is stabilised by an intramolecular hydrogen bond (K13-S28) and a dense network of intermolecular contacts with the ribosome. Specifically, the RBD binds the 28S rRNA's helices 47, 57, and 59 and ribosomal proteins eL19, 22, and 31, via electrostatic interactions and via hydrophobic interactions with pockets on each of the three ribosomal proteins. The only other factor known to bind in this region is the nascent polypeptide-associated complex subunit β (NACβ) (*Jomaa et al., 2022*), whose anchor domain has a very different structure but would partly clash with the RAMP4 RBD.

RAMP4's RBD is connected by a flexible linker to the kinked TMD. This region is less well resolved, so to inform modelling we used AF2 to predict the structure of the Sec61•RAMP4 complex. The resulting prediction is confident (*Figure 3—figure supplement 1A*) and agrees with the density map. To check that this prediction was specific to RAMP4, we used AF2 to screen a diverse panel of other TMDs and SPs, and found that RAMP4 was indeed the only protein predicted to bind Sec61 (*Figure 3—figure supplement 1B*). Thus, our model of the RAMP4 TMD is supported by both the density map and a confident, specific structure prediction.

The cytoplasmic half of the RAMP4 TMD is hydrophobic and binds the open Sec61 gate as if it were a TMD or SP (*Figure 3C*). Bound here, RAMP4 holds the Sec61 pore ring wide, just as SPs and TMDs do (*Figure 2C*). Unlike an SP or TMD, however, RAMP4 does not displace the plug helix. Instead, the plug moves together with the widening pore ring, keeping it plugged. This observation contrasts with prior speculation that pore widening is sufficient to trigger plug displacement (*Voorhees and Hegde, 2016a*). Pore widening and unplugging do indeed occur together when the gate is opened by an SP or TMD (*Figure 2E*), but RAMP4 shows that widening can occur without unplugging. This implies that unplugging occurs when an SP or TMD pulls a flanking segment of nascent chain through the channel pore, and this necessarily displaces the plug. RAMP4, being a tail-anchored protein, threads nothing through the pore, and thus does not clash with the plug.

At its midpoint, the TMD of RAMP4 is kinked 40° at a conserved glycine, and the lumenal half of the TMD is amphipathic. The TMD's hydrophilic face is oriented towards the channel interior and

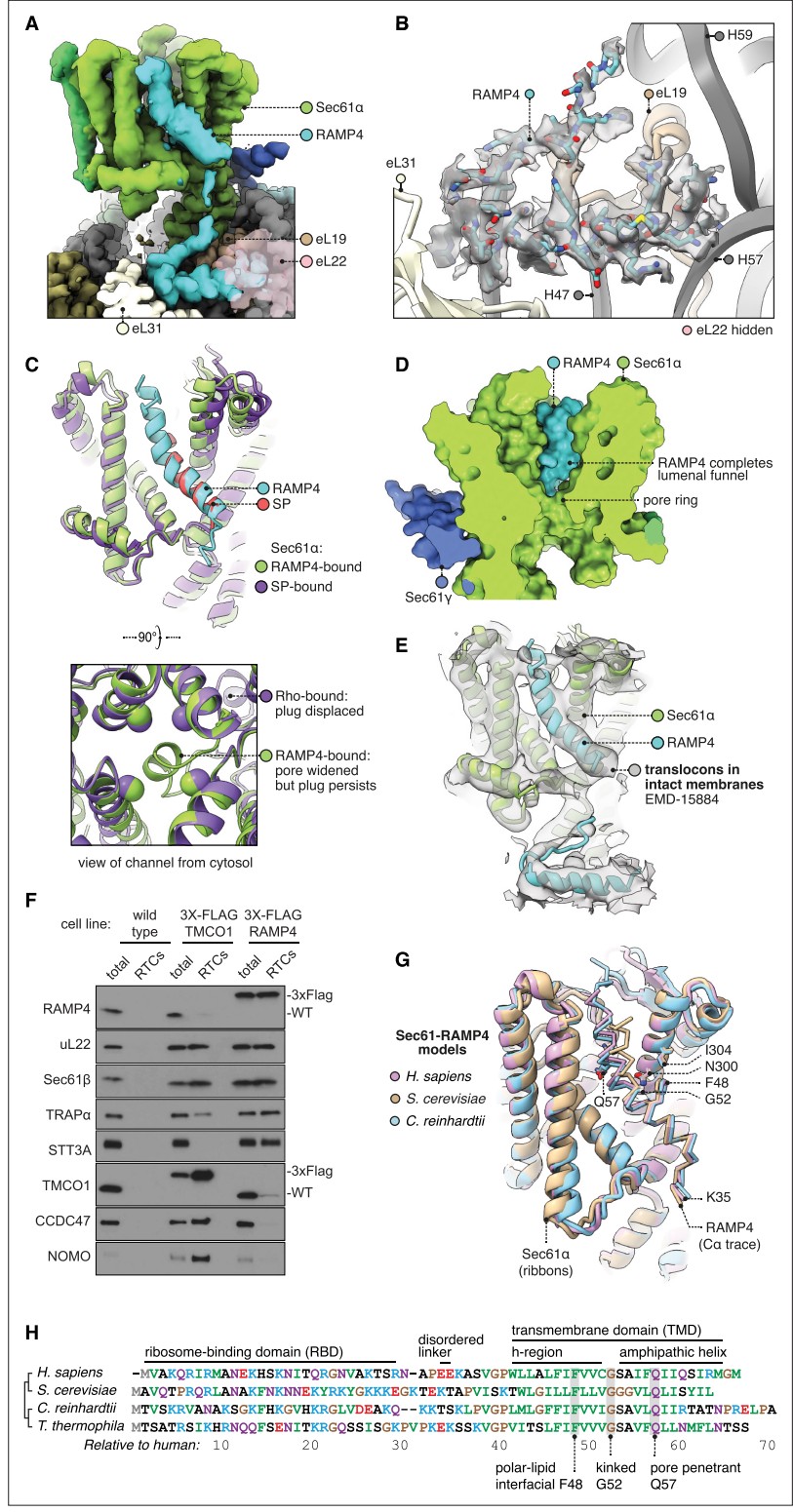

**Figure 3.** Structure of Sec61 bound to RAMP4. (**A**) Overview of the RAMP4-bound Sec61 complex. For display, the density map was filtered using DeepEMhancer and clipped along a plane adjacent to RAMP4. eL22 is shown at 70% opacity to avoid occluding RAMP4. (**B**) The RAMP4 ribosome-binding domain (RBD) fit to density. For display, the density map was supersampled at half its original pixel size. eL22 is hidden to avoid occluding RAMP4. (**C**) RAMP4-bound Sec61 is structurally similar to signal peptide (SP)-bound or transmembrane domain (TMD)-bound Sec61, except RAMP4 does not displace the plug helix, whereas SPs and TMDs do. The SP is

*Figure 3 continued on next page*

*Figure 3 continued*

preprolactin (PDB 3JC2 rebuilt into EMD-3245 using restraints from AF2). (**D**) RAMP4 contributes to the lumenal funnel of Sec61. (**E**) Alignment of the RAMP4-Sec61 model to a cryo-ET map of the Sec61-TRAP-OSTA translocon in intact membranes (EMD-15870; *Gemmer et al., 2023b*). (**F**) Immunoblotting analysis of ribosome-translocon complexes (RTCs) affinity purified from the indicated HEK293 cell lines. 1% of total microsomes isolated from the cells ('total') is shown for comparison. Note that expression levels of 3X-FLAG-tagged TMCO1 and RAMP4 are comparable to their native levels seen in wild-type cells. Similar results were observed in two biological replicates. (**G**) Superposition of AF2-predicted RAMP4-Sec61 structures from different species. (**H**) Alignment of select RAMP4 sequences.

The online version of this article includes the following source data and figure supplement(s) for figure 3:

**Source data 1.** Uncropped full gel images of the immunoblots shown in *Figure 3F*.

**Figure supplement 1.** Predicted structures of Sec61-RAMP4 complexes.

**Figure supplement 2.** RAMP4 occupancy in different maps.

completes the hydrophilic lumenal funnel of Sec61α (*Figure 3D*). RAMP4's integral contribution to forming the Sec61 channel explains why it, and not the more peripheral Sec61β or γ subunits, cross-links with nascent chain-encoded photoprobes in the Sec61 channel (*Schröder et al., 1999*).

Comparing different classes of particles, we see that RAMP4 competes with other factors. RAMP4 is partially or completely depleted from classes in which the gate is closed, occupied by RhoTM2, or latched shut by the PAT complex (*Figure 3—figure supplement 2*). By competing with RAMP4's TMD, RhoTM2 and PAT also reduce the occupancy of its RBD, which indicates that the RBD alone binds too weakly to be retained during extraction, purification, and grid preparation. This is consistent with biochemical evidence that RAMP4's association is detergent-sensitive (*Görlich and Rapoport, 1993*). Together, these factors explain why RAMP4's occupancy in prior cryo-EM maps was low enough to be overlooked, although in hindsight it is visible in several of them (*Braunger et al., 2018*; *Voorhees et al., 2014*; *Pfeffer et al., 2015*).

To assess the abundance of Sec61•RAMP4 complexes in native membranes, we examined the best native maps currently available, from the subtomogram averages of ribosomes from human cell-derived microsomes (*Gemmer et al., 2023b*). This dataset was separated into four classes of RTCs: 70% Sec61•TRAP•OSTA, 12% Sec61•MPT, 10% Sec61•TRAP, and 9% Sec61•TRAP•MPT. Examining each class average for the RAMP4 RBD, we find that it is present in the classes without MPT and absent from the classes with MPT, consistent with our single-particle analysis. We then estimated the occupancy of RAMP4 using OccuPy (*Forsberg et al., 2023*), and found that it is present in ~85% of Sec61•TRAP•OSTA RTCs and ~53% of Sec61•TRAP RTCs (*Figure 3—figure supplement 2*), which equals ~81% of the non-MPT RTCs. Thus, RAMP4 is absent from MPT-containing RTCs and seems to be present in almost all non-MPT RTCs.

Alongside the RAMP4 RBD, the subtomogram averages also show density for RAMP4's kinked TMD (*Figure 3E*). Previously, this TMD density had been speculated to represent SPs (*Braunger et al., 2018*; *Pfeffer et al., 2015*; *Voorhees and Hegde, 2016b*). But the helical density observed is ~25 aa long, like RAMP4's TMD, whereas the helices of SPs are only 7–15 aa long (*von Heijne, 1985*). More-over, the occupancy of this TMD is high, even after hours of translation inhibition (*Figure 3—figure supplement 2*; *Gemmer et al., 2023b*), when SP occupancy should be low due to their co-translational dissociation and subsequent processing and degradation (*Liaci et al., 2021*; *Jungnickel and Rapoport, 1995*; *Mothes et al., 1994*; *Lyko et al., 1995*; *Lemberg and Martoglio, 2002*). We therefore assign the kinked TMD density in these earlier maps to RAMP4, implying that most co-translational translocation normally occurs through Sec61•RAMP4 channels.

Biochemical analyses provided independent support for the structural observation that RAMP4 and MPT compete for occupancy at the ribosome-Sec61 complex (*Figure 3F*). In this experiment, HEK293 cells knocked out for either TMCO1 (of the GEL complex) or RAMP4 were engineered to re-express a tagged version at near-native levels. Microsomes from these cells were then used to affinity-purify RTCs via the tagged protein, then immunoblotted for various translocon-associated factors. The results show that RAMP4-purified RTCs contain little or no MPT subcomplexes, but do contain OST and TRAP. By contrast, TMCO1-purified RTCs contain very little RAMP4 and OST, but the full complement of MPT subcomplexes and slightly diminished TRAP. Thus, RAMP4- and MPT-containing RTCs are mostly mutually exclusive, with the former linked to co-translational translocation

through the Sec61 channel and the latter linked to membrane protein insertion at the hinge side of a closed Sec61 channel.

Having described the structure of mammalian RAMP4, we briefly consider how widely conserved this structure may be. Examining representative model organisms (*S. cerevisiae* [*Sc*] and *C. reinhardtii* [*Cr*]), we find that in both cases RAMP4 is confidently predicted to bind Sec61 like animal RAMP4 does (*Figure 3—figure supplement 1A*). Especially conserved is the region surrounding RAMP4's glycine kink and Sec61 N300 (*Figure 3G and H*), which is part of Sec61's polar cluster that is important for gating (*Trueman et al., 2012*). To assess when RAMP4 arose in evolution, we searched for homologs in several early-branching eukaryotic taxa (Discoba, Metamonada, Malawimonada) and found it to be largely absent from those groups, suggesting an origin more recent than the last eukaryotic common ancestor. Thus RAMP4, although not ubiquitous, is present across the major eukaryotic kingdoms and forms a dynamically associating part of the Sec61 channel.

## uL22's C-tail switches to contact Sec61 and the nascent chain

In our non-MPT maps, we were surprised to observe density for the C-tail of ribosomal protein uL22 (*Figure 4A*). This tail has not been described in any prior structural studies. Here, we find it stretched across the ribosome's membrane-facing surface towards Sec61 and the nascent chain, contacting eL31 and several RNA helices along the way (*Figure 4A*). Density for the C-terminal helix (CTH) is moderately strong (*Figure 1—figure supplement 2A*) and similar in each class of particles, indicating that it is not strongly correlated with TRAP or RAMP4 occupancy, nor Sec61 conformation or nascent chain binding to Sec61. The sole exception is the absence of the CTH from the MPT map (*Smalinskaitė et al., 2022*), where it would clash with the gate latch helices of the PAT complex (*Figure 4A*). It would also clash with SRP (via SRP54's M-domain) (*Kobayashi et al., 2018*; *Voorhees and Hegde, 2015*; *Jomaa et al., 2021*), NAC (via its ribosome-binding helices) (*Jomaa et al., 2022*), RAC (via Zuo1) (*Kišonaitė et al., 2023*), NatA (via Nat1) (*Knorr et al., 2019*), and NatB (via MDM20) (*Knorr et al., 2023*).

Having seen that uL22 can adopt an engaged or displaced state in different RTCs, we then compared these states to a map of cytoplasmic ribosomes without bound translocons (EMD-40205, the best-resolved mammalian map available) (*Holm et al., 2023*). The cytoplasmic ribosome displays clear density for the uL22 tail up to the point where it binds H24/47, but the following CTH is diffuse, indicating that it is flexibly anchored to H24/47. Anchored here, the CTH would be well positioned to scan emerging nascent chains and sense when Sec61 binds.

To judge how common uL22 engagement is, we re-examined publicly available RTC maps for previously overlooked uL22 density. We found clear uL22 CTH density in maps from several recent studies (*Figure 4—figure supplement 1*; *Gemmer et al., 2023b*; *Jaskolowski et al., 2023*; *Pauwels et al., 2023*). Most importantly, it is present in subtomogram averages of RTCs transporting diverse endogenous nascent chains through intact microsomal membranes (*Gemmer et al., 2023b*), and the uL22 CTH density in those maps is just as strong as in the present dataset (~40% occupancy; *Figure 4—figure supplement 1A*). This indicates that uL22 engagement is common in native translocons. It is unclear why it is not visible in earlier maps; it may be present and not well resolved, or absent due to a preference for specific nascent chain features.

Comparing the uL22 tail across species, we find that it is conserved by the earliest branches of the animal tree, but not by the nearest non-animal branches (*Figure 4B*). Although fungi have an elongated uL22 tail, it is unlike the animal tail in sequence, and appears to have arisen independently. Indeed structures show that the fungal uL22 tail binds to an entirely different site (*Figure 4C*; *Best et al., 2023*), and does so constitutively instead of dynamically. Taken together, this suggests that the animal uL22 tail structure was acquired during the evolution of the first animals. Among animals, the uL22 tail's conservation suggests that it is functionally important. Particularly well conserved is the SXKK motif that initiates the CTH and binds H24/47 (*Figure 4D*). The tip of the CTH that would contact nascent chains typically contains a mix of basic, acidic, and hydrophobic residues, and this complexity makes it difficult to predict what if any nascent chain features it may recognise.

It is noteworthy that when engaged, the uL22 CTH blocks a gap between the ribosome and Sec61 that would otherwise allow the nascent chain to exit the channel vestibule and enter the cytoplasm (*Figure 4E*). This gate-side exit is one of two such exits, with the other being on the Sec61 hinge side, where multipass proteins exit for insertion by the MPT (*Smalinskaitė et al., 2022*). While uL22 blocks

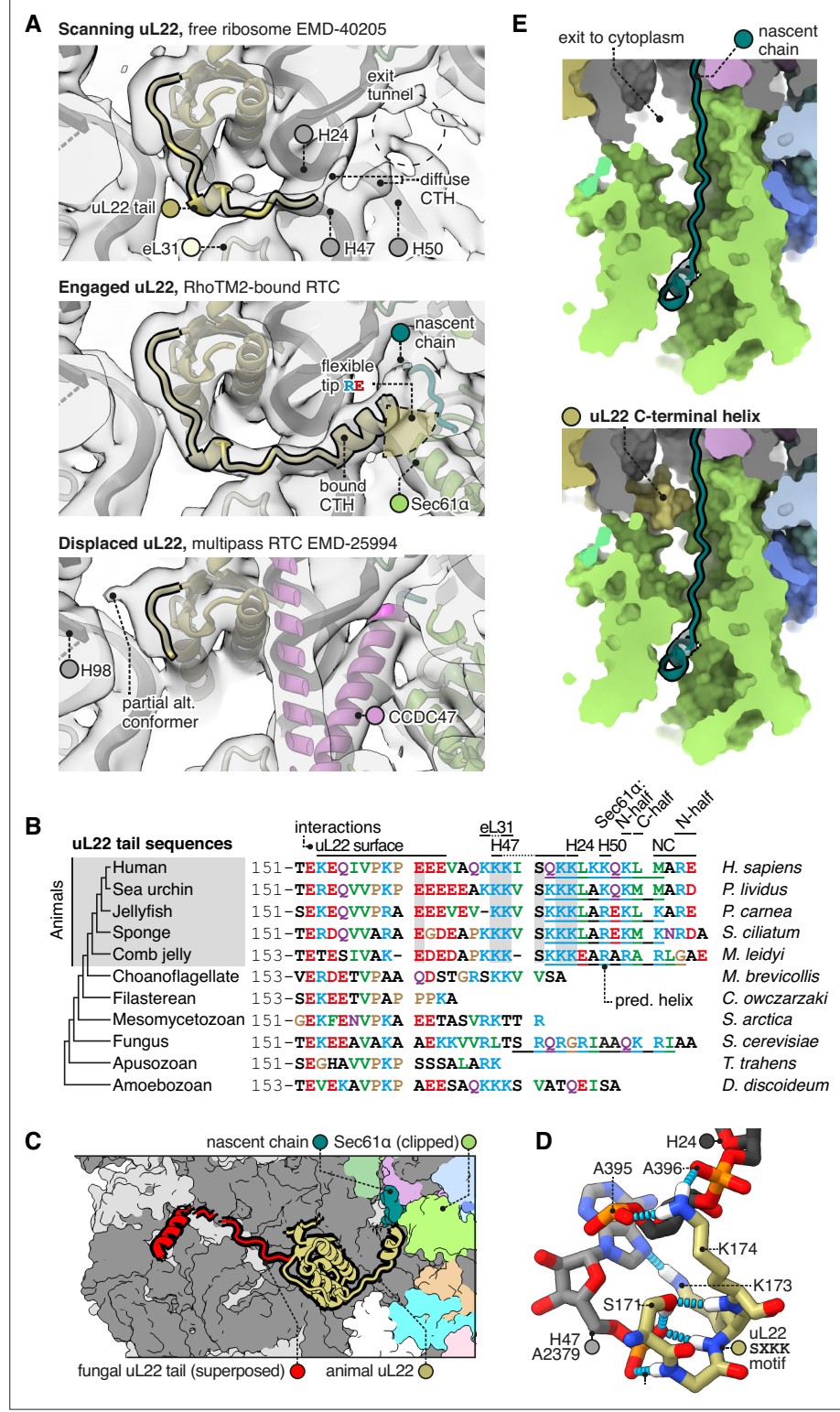

**Figure 4.** The C-terminal helix of animal uL22 binds the ribosome-translocon junction. (**A**) uL22 has three discernible states: scanning, engaged, or displaced. The maps shown are as follows: for the scanning state, the highest-resolution map available of a cytoplasmic ribosome from eukarya (EMD-40205); for the engaged state, the RAMP4-bound Sec61 map from this study, and for the displaced state, the multipass translocon (MPT)-bound map previously reported from this dataset (EMD-25994). For display, the maps were lowpass-filtered at 8 Å. (**B**) Alignment of select uL22 tail sequences. Underlines indicate the C-terminal helices annotated in the AF2

*Figure 4 continued on next page*

*Figure 4 continued*

database. (**C**) Superposition of the animal and fungal uL22 tails (PDB 8AGX). (**D**) Structure of the uL22 SXKK motif, with hydrogen bonds indicated in cyan. (**E**) Comparison of the ribosome-translocon junction in the absence (top) and presence (bottom) of an ordered uL22 C-terminal helix. Note that the helix occludes the gate-side exit towards the cytoplasm.

The online version of this article includes the following figure supplement(s) for figure 4:

**Figure supplement 1.** Occupancy of the uL22 C-terminal helix (CTH) in different maps.

the gate-side exit, the hinge-side exit would remain open. The functional consequences of this change are currently unclear.

## Structure of the TRAP complex

The occupancy of the hetero-tetrameric TRAP complex (comprised of α, β, γ, and δ subunits) in the all-particle map was low, so we performed focused classification on this region and obtained two well-resolved TRAP classes (*Figure 1—figure supplement 1C*). TRAP in these two classes adopts slightly different tilts with respect to the ribosome, which we call conformations 1 and 2, although the true range of TRAP conformations is presumably continuous rather than discrete. The TRAP maps were well fit by a predicted model (*Figure 5A*) after minor adjustments. One additional protein density was observed in TRAP class 1, consisting of a TMD and cytoplasmic helix bound to TRAPγ. We provisionally assign this density to Calnexin because it is by far the major TRAP-binding protein (*Fons et al., 2003*; *Wada et al., 1991*) and it is known to bind TRAP via its TMD. The putative TRAP-Calnexin interaction was not detected by AF2, perhaps because Calnexin di-palmitoylation, which AF2 ignores, is crucial for TRAP interaction with Calnexin (*Lakkaraju et al., 2012*). Anchoring of Calnexin at this position would explain how its elongated lumenal domain interacts co-translationally with nascent chains as they are translocated by RTCs (*Molinari and Helenius, 2000*; *Daniels et al., 2003*).

The TMDs of TRAPβγδ and the provisionally assigned Calnexin all bundle together, but not the TMD of TRAPα. Instead, the TRAPα TMD binds the lumenal hinge loop of Sec61α, as does the TRAPα lumenal domain (*Figure 5B.3*). While making those lumenal contacts, the TRAPα TMD tilts its cytoplasmic end away from Sec61α, allowing conserved basic residues flanking its TMD to contact the tip of 5.8S rRNA helix 7 (*Figure 5B.4*). Strikingly, this rRNA segment has undergone a dramatic rearrangement from its cytoplasmic state, which puts C81, U85, and U86 in contact with the membrane and the basic residues in TRAP (*Figure 5B.5*). From there, the C-tail of TRAPα continues along the ribosome surface and binds tightly at a site contacting uL23, uL29, and 5.8S rRNA helix 9 (*Figure 5B.6*). Thus TRAPα makes extensive contacts with both the ribosome and Sec61.

Besides TRAPα, TRAPβ, TRAPγ, and Calnexin also contact the ribosome and Sec61: TRAPβ's basic C-terminal residues contact the 5.8S rRNA at H9 ES3 (*Figure 5B.7*); TRAPγ's helical hairpin residues R110, K111, and especially R114 contact the 28S rRNA at H54 and ES26 (*Figure 5B.8*); TRAPγ's flexible N-tail helix binds eL38 (*Figure 5B.9*); the provisionally assigned Calnexin's flexible C-tail contacts the 28S rRNA at H63 ES27a (*Figure 5B.10*); TRAPγ's C-terminus contacts Sec61γ's N-terminus (*Figure 5B.1*); and TRAPγ's lumenal loop 3/4 contacts Sec61α's lumenal loop 8/9 (*Figure 5B.2*). Altogether, TRAP's multivalent site-specific and non-specific interactions across many sites via both rigid and flexible elements explain how it remains associated with the translocon while adopting a range of positions and orientations, compared to other translocation factors like OST or the MPT which bind at relatively fixed sites.

## TRAP competes and cooperates with different translocon subunits

The TRAPδ Ig-like domain contains a helical hairpin between β-strands 6 and 7. No similar hairpin appears in any other domain in the AFDB50 database queried by Foldseek (*van Kempen et al., 2024*; *Figure 6—figure supplement 1A*). This unique hairpin is isolated in our Sec61-TRAP structure, but in Sec61-TRAP-OSTA structures (*Gemmer et al., 2023b*), it extends towards OST-A subunit RPN2 (*Figure 6A*). A basic patch on TRAPδ is separated by less than 5 Å from an acidic patch on RPN2, indicating that they would share an electrostatic attraction (*Figure 6A*, inset). Each partner presents three conserved charges, and the closest pair (K117 on TRAPδ and D386 on RPN2) is the most conserved, indicating that their attraction is functionally important. OSTA's attraction to TRAPδ is weak compared to its binding to the ribosome, but TRAPδ may nonetheless help recruit OSTA, since TRAPδ

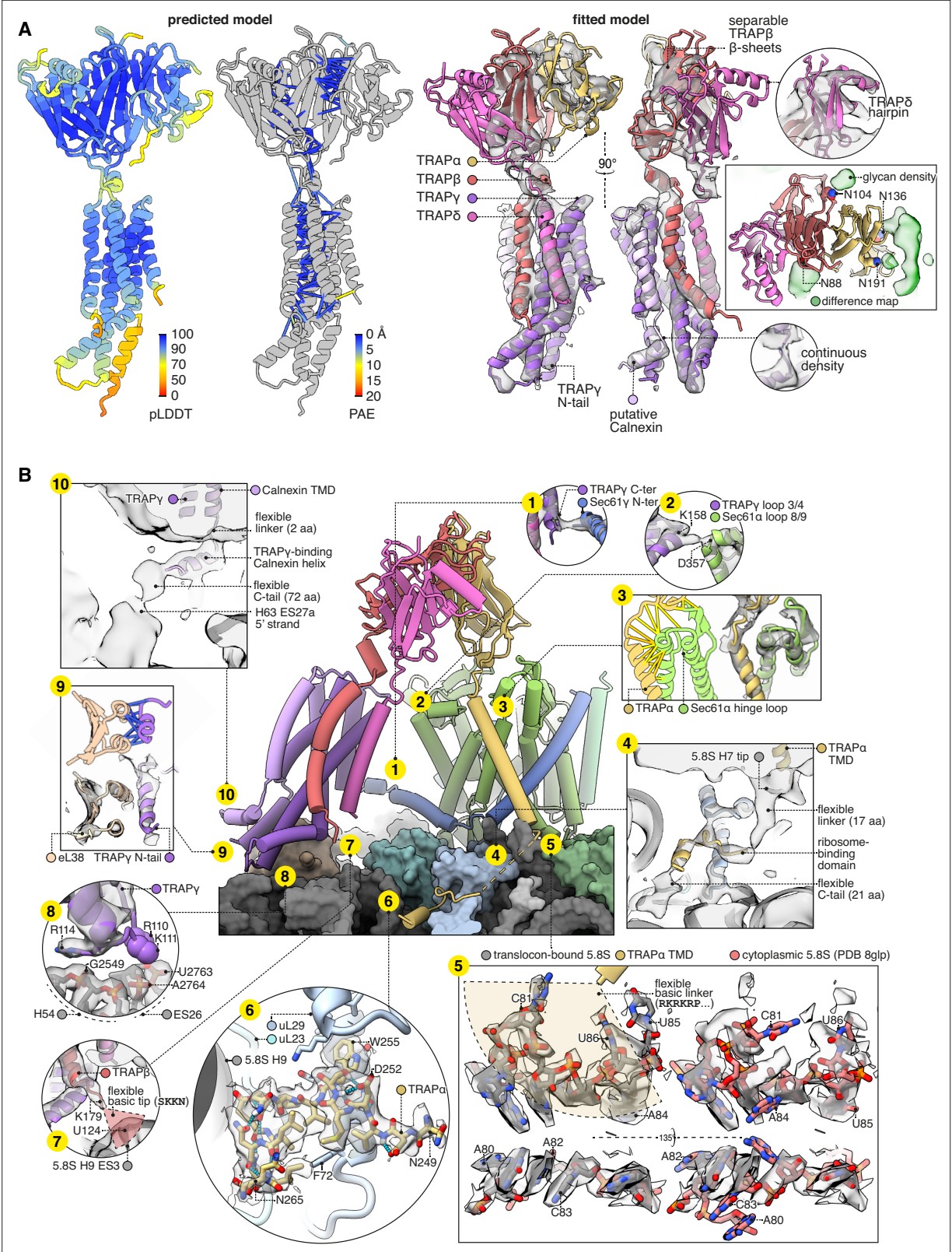

**Figure 5.** Structure of the translocon-associated protein (TRAP) complex and its contacts with the ribosome and Sec61. (**A**) The predicted structure of the TRAP complex (left) fitted into the EM density shown with a semitransparent isosurface (right). For display, the density map was lowpass-filtered at 8 Å. The structural model at left is coloured by the predicted local distance difference test (pLDDT). Colour-coded pseudobonds indicating the predicted aligned error (PAE) between pairs of residues are shown at subunit interfaces, and reflect the confidence with which AF2 predicts intersubunit

*Figure 5 continued on next page*

Figure 5 continued

contacts. In the density-fitted structure, the chain putatively assigned to Calnexin is also shown, where it was fitted to an additional density on the surface of TRAPγ. (**B**) TRAP's contacts with the ribosome and Sec61. The 5.8S rRNA in subpanel 5 is shown in two conformations, both fit to the same density map, to illustrate the conformational change induced by association with the translocon. For display, the density map in subpanels 1–4, 7, 9, and 10 was lowpass-filtered at 8 Å; in subpanel 6, the map was supersampled at half the original pixel size; in subpanel 8, the map was filtered by DeepEMhancer. Models shown in flat lighting are AF2 predictions with pseudobonds colour-coded by PAE as in panel (**A**).

would attract OSTA from most possible angles of approach, whereas OSTA's ribosome contacts are stereospecific. The TRAPδ-OSTA interaction may explain why TRAPδ defects cause congenital disorders of glycosylation (*Losfeld et al., 2014*; *Ng et al., 2019*; *Phoomak et al., 2021*).

Comparing the TRAP structure to the MPT structure (*Smalinskaitė et al., 2022*), we find that TRAP would clash with part of the MPT, namely the BOS complex comprised of TMEM147, Nicalin, and NOMO. The prior BOS model omitted NOMO, so to fully characterise this clash we ran a prediction of the full BOS complex structure, obtaining a high-confidence prediction that fits a previously reported subtomogram average (*Braunger et al., 2018*; *Figure 6B*). As an aside, it is noteworthy that the first prealbumin-like domain (PLD1) in NOMO is predicted to bind PLD10 and PLD11, suggesting that its reported ability to regulate the spacing between ER sheets (*Amaya et al., 2021*) results from it forming antiparallel homodimers across the ER lumen (*Figure 6—figure supplement 1B*). Contemporaneous work (*Gemmer et al., 2023a*) has arrived at a similar model for PLD10-12 but did not model PLD1.

Comparing the TRAP and BOS structures, TRAPα competes with BOS subunit TMEM147 for the same binding sites on the hinge loop of Sec61α and the 5.8S rRNA's helix 7, while PLD12 of NOMO would clash to a more limited degree with the lumenal domain of TRAPβ. When TRAP is displaced by BOS, its interactions with Sec61 are disrupted and its transmembrane bundle (which is no longer visible in the cryo-EM map) evidently pivots towards the still-bound TRAPα C-tail RBD, causing a pronounced shift in the shape of the detergent micelle (*Figure 6C*). This extensive competition resulting in partial TRAP displacement explains why prior studies suggested that TRAP is present in only 40% of MPT complexes, but at high occupancy at all other RTCs (*Gemmer et al., 2023b*; *Gemmer et al., 2023a*). However, the high occupancy of the TRAPα C-tail RBD in MPT-containing translocons suggests that TRAP does not fully dissociate upon MPT recruitment and remains in vicinity of the translocation site.

## Additional functionally relevant TRAP features

The above analysis shows that TRAP binds together ribosomes, OSTA, and Sec61, but competes with the MPT, whose presence inhibits Sec61. In principle, these activities could suffice to explain TRAP's observed functions as a stimulator of Sec61-dependent secretion (*Fons et al., 2003*) and OSTA-dependent glycosylation (*Phoomak et al., 2021*). However, we observe four additional TRAP features that appear functionally relevant and are therefore noteworthy.

First, TRAPα presents a conserved patch to the nascent chain where it first emerges from Sec61 (*Figure 7A*). This patch could potentially bind the nascent chain or redirect it towards the active site of OSTA. Second, by binding the C-half of Sec61, TRAP can potentially influence the opening of the lateral gate (*Figure 7B*). If TRAP caused the C-half to favour opening, this would explain in part why it stimulates the recognition of weakly gating signals (*Fons et al., 2003*). The combined effect could explain its overall stimulatory role in translocation.

Third, TRAP prefers a membrane plane tilted 20° relative to Sec61's, and imposes this curvature on the surrounding micelle (*Figure 7C*). The same tilt is observed in Sec61-TRAP maps from intact membranes, but it is unclear whether TRAP imposed this curvature on the membrane, since it is the same degree of curvature found in native ER tubes and sheet edges (*Shibata et al., 2010*). Thus, while it is possible that TRAP modulates Sec61 activity by disrupting the local membrane, it may instead have adapted to reside in and sense pre-existing membrane deformations.

Fourth, the lumenal domain of TRAPα near the Sec61 channel has an extraordinarily long and acidic N-tail tipped by hydrophobic residues (*Figure 7D*). This could interact with any lumenal part of the translocon or nascent chain. No data is available on this tail's function, aside from the fact that this and Calnexin each bind far more calcium than any other protein in ER membrane extracts (*Wada et al., 1991*) as expected for its exceptional charge and abundance.

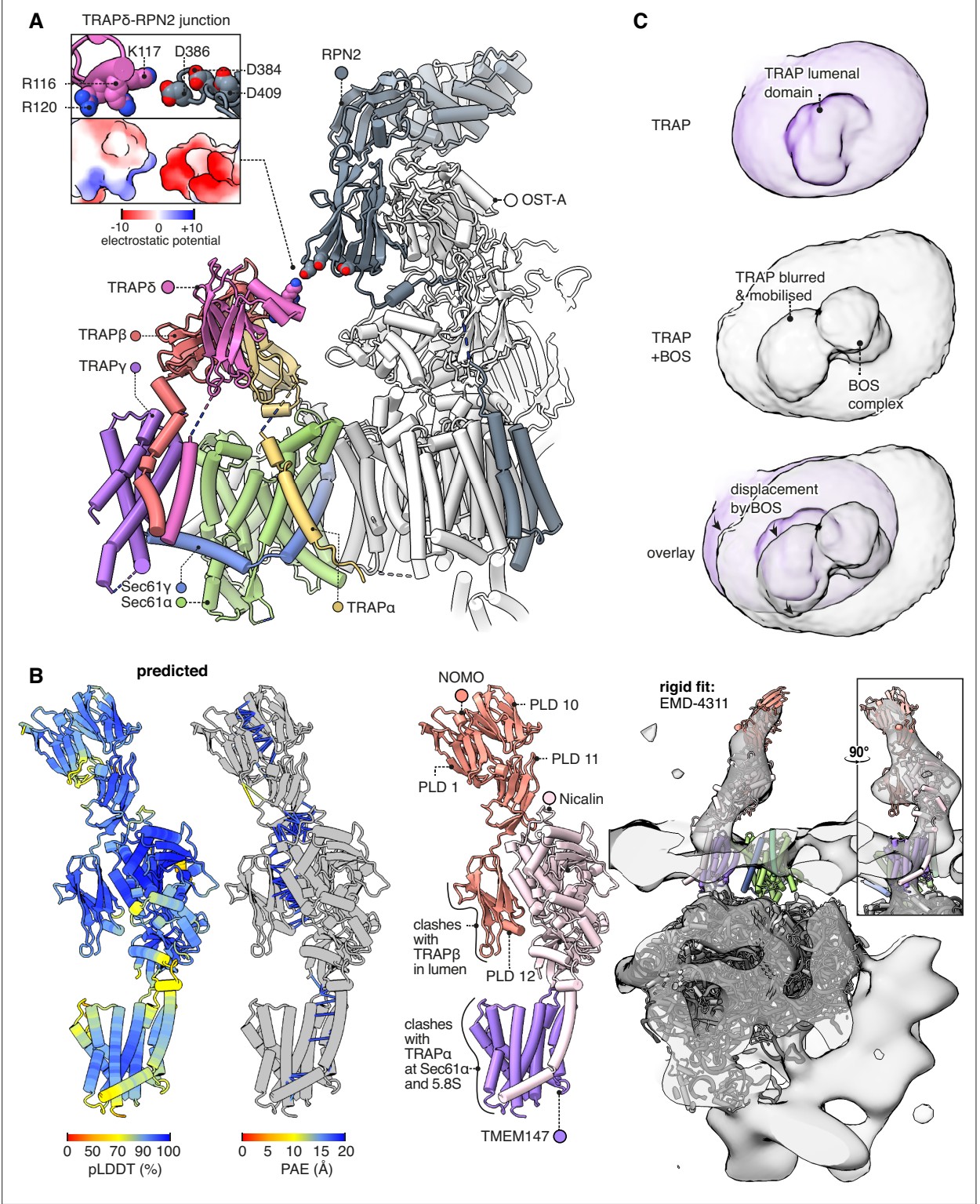

**Figure 6.** Interactions between the translocon-associated protein (TRAP) complex and other translocon constituents. (**A**) Overview of the Sec61-TRAP-OSTA complex (PDB 8B6L; *Gemmer et al., 2023b*). The individual subunits of OSTA are not labelled or coloured separately, except for Ribophorin II (RPN2). The inset shows how the basic hairpin on TRAPδ is close enough to interact electrostatically with acidic loops on RPN2. The indicated amino acids are highly conserved in both TRAPδ and RPN2. Because the rotameric states of K117 and D386 are uncertain, the rotamers yielding the smallest gap are shown. (**B**) The predicted structure of the BOS complex is shown in flat lighting at left, and a rigid-body fit to EM density is shown with a semitransparent isosurface at right. PLD stands for prealbumin-like domain. (**C**) BOS complex destabilises TRAP's association with the translocon. The

*Figure 6 continued on next page*

*Figure 6 continued*

densities shown were Gaussian-filtered with a B-factor of 2000. The BOS+TRAP map is the multipass translocon (MPT) map previously reported from this dataset (EMD-25994).

The online version of this article includes the following figure supplement(s) for figure 6:

**Figure supplement 1.** Additional analysis of the translocon-associated protein (TRAP) and BOS complexes.

## TRAP conservation across eukaryotes and archaea

Having described the features of metazoan TRAP, here we briefly contrast it with TRAP in representative model organisms from plantae (*Cr*), fungi (*Sc*), and excavates (*Trypanosoma brucei* [*Tb*]), a group of early-branching eukaryotes. *Cr* has already been shown by subtomogram averaging to contain TRAPαβ-like density at its translocons (*Pfeffer et al., 2017*). Indeed we find that its genome contains TRAPαβ but not γδ, and the predicted structure of *Cr* TRAPαβ-Sec61 fits the tomographic density with minor adjustments (*Figure 8A*). *Cr* TRAP conserves animal TRAP's predicted binding to the Sec61 hinge and its C-tail RBD. However, whereas animal TRAP binds to the ribosome at eL38 via TRAPγ, *Cr* TRAP is predicted to bind this same ribosomal protein via TRAPβ (*Figure 8A*, inset). Thus, the *Cr* and animal complexes share the similar ribosome-binding sites despite their differences in composition. As an aside, it is noteworthy that the *Cr* TRAPβ TMD is too hydrophilic to insert on its own ($\Delta G_{pred}$ = 1.824, i.e., 5% insertion), suggesting that it is bound and stabilised by an unidentified protein functionally analogous to TRAPγ.

*Sc* has no annotated TRAP genes, and indeed it has been suggested that most fungi lack TRAP (*Pfeffer et al., 2017*). However, we find that *Sc* contains a previously unidentified gene for TRAPα (Irc22, HHpred $E=10^{-35}$). This suggests TRAP's prevalence in fungi has been underestimated. *Sc* TRAPα conserves animal TRAPα's predicted binding to the Sec61 hinge and its C-tail RBD (*Figure 8B*). No other TRAP subunits were detected in *cerevisiae* by sequence- or structure-based searches (HHpred, Foldseek). Thus TRAP's links to co-translational translocation may be conserved in fungi, but its complexity is dramatically reduced.

The early-branching excavate *Tb* has TRAPα, β, and γ, but no δ (*Figure 8C*). *Tb* TRAPαγ conserve the same predicted interactions as their homologs in fungi, plants, and animals (where found). Overall, the observed distribution and conservation of features suggest that the original eukaryotic TRAP consisted of TRAPαβγ, bound the Sec61 hinge loop via TRAPα, bound the ribosome flexibly via the TRAPα C-tail RBD and the TRAPγ N-tail, and also contacted the ribosome via the TRAPβγ bundle.

If TRAPαβγ were present in the last eukaryotic common ancestor, they may have been inherited from archaea, which are the ancestors of eukaryotes. Structural queries of eukaryotes' archaeal sister group (Heimdallarchaeota) identified candidate homologs of TRAPα ($E=9.95 \times 10^{-8}$), β ($E=5.15 \times 10^{-10}$), possibly TRAPγ ($E=1.84$), and not TRAPδ, as expected. To test whether these candidates are also similar to TRAPαβγ in sequence, we used them to perform reciprocal HHpred queries of the human proteome, and in each case the corresponding human TRAP protein was the top hit ($E=0.031$ for TRAPα, $9.4\times10^{-14}$ for TRAP β, and 110 for TRAPγ). A contemporary study has also proposed TRAP homologs in Heimdallarchaeota (*Eme et al., 2023*), although caution is warranted in some of these assignments because many do not share predicted structural similarity to TRAP subunits or sequence similarity with human TRAP in reciprocal HHpred searches.

We then queried AF2 to see whether the top-scoring archaeal TRAPαβγ hits were predicted to form a complex with each other or with SecY. No contacts among the top hits were detected, but this could be an artefact of how few sequences AF2 could collect to constrain the prediction (as few as 5). By contrast, contacts were confidently predicted between archaeal TRAPα and SecY, forming a dimer similar to the eukaryotic TRAPα-Sec61 dimer (*Figure 8D*). This similarity was predicted even though no eukaryotic TRAPα were included in the input alignment or training set, and thus it was based on archaeal TRAPα alone.

While performing sequence searches with TRAPα and β, we were surprised to find that they are much more similar in sequence than we had expected (HHpred $p=2 \times 10^{-7}$), suggesting that they share a common ancestor. In fact they are more similar to one another than to any other human proteins (HHpred), including other Ig-like domains. By contrast, TRAPδ is quite dissimilar (HHpred α $p=0.64$, β 0.66). Thus, although TRAP's three lumenal domains all have Ig-like folds, it appears that two of them, TRAPα and β, originated when an archaeal proto-TRAP protein duplicated, whereas TRAPδ is

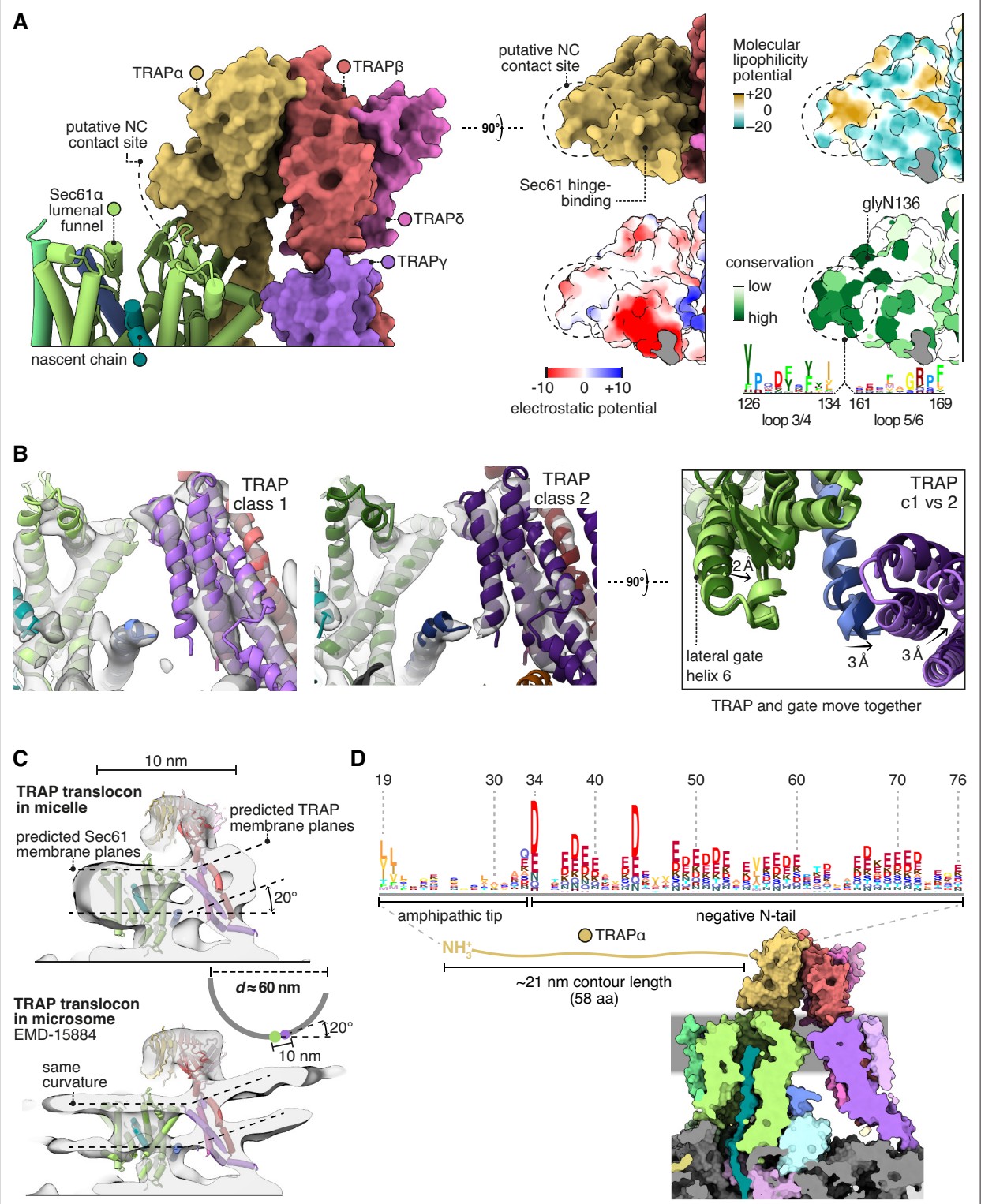

**Figure 7.** Translocon-associated protein (TRAP) features that may influence Sec61 activity. (**A**) The region of TRAPα closest to the lumenal vestibule of Sec61 represents a putative contact site with translocating nascent chains. This region is highly conserved. (**B**) Image classification separated two slightly shifted TRAP conformations, shown in their respective densities in the left and middle panels. The superposition of the two models shows that this shift in TRAP correlates with a shift in the C-half of Sec61's lateral gate (helix 6). The helices coloured in blue are Sec61γ, which moves together with TRAPγ (purple). For display, the density maps were lowpass-filtered at 8 Å. (**C**) Top: The TRAP-Sec61 complex induces curvature in detergent micelles, consistent with the membrane planes predicted by the Orientation of Proteins in Membranes (OPM) server. Bottom: The structure of the TRAP-Sec61

*Figure 7 continued on next page*

Figure 7 continued

complex in microsome membranes shows the same curvature as observed in detergent micelles. Inset: The radius of curvature induced in the micelle is ~30 nm (diameter 60 nm), which matches that of endoplasmic reticulum (ER) tubules and sheet edges. (**D**) TRAPα's N-tail is anionic, amphipathic, and is positioned to interact with Sec61, nascent chains, or other factors. The N-tail is shown to scale. Note the conservation of the amphipathic tip and anionic character. The logo plots in panels (A and D) represent an HMM generated by jackHMMER upon convergence after querying UniProtKB's metazoan sequences with the human TRAPα sequence. Only signal above background is shown, as rendered by http://skylign.org/.

evolutionarily unrelated (*Figure 8E*). If proto-TRAP formed dimers, it would presumably resemble a homomeric version of the heterodimeric TRAPαβ complex found in plants (*Pfeffer et al., 2017*).

Towards understanding TRAP's mechanism, it is noteworthy that although its affinity for the ribosome and Sec61 are conserved, at least two of the four other functionally relevant features highlighted above are not conserved: yeast TRAP's single TMD is unlikely to deform membranes, and TRAP from many organisms does not have a polyacidic lumenal domain. The other two features, allosteric effects on gating and a putative substrate-binding activity, could plausibly be conserved together with the TRAPα-Sec61 structure.

## Conclusions and perspective

The most striking finding of this work is that the Sec61 protein secretion channel essentially has a fourth subunit, RAMP4, which is present in about 80% of non-MPT RTCs. Our structure shows that RAMP4 binds to Sec61 like an SP, thereby blocking the channel's lateral gate, holding wide its central pore, and completing its lumenal funnel. For these reasons, we speculate that RAMP4 acts as a surrogate SP. It is possible that once a secretory protein's SP dissociates from the lateral gate, the central channel would narrow, thereby impeding translocation speed or efficiency. By binding the lateral gate during these later stages of translocation, RAMP4 could smooth transport through the channel for certain types of sequences. Such a role would explain why RAMP4 seems to be present during the vast majority of co-translational translocation through Sec61. Although RAMP4 appears to be eukaryote-specific, other proteins may serve similar functions in some prokaryotes. For example, the first TMD of *Escherichia coli* YidC invades the lateral gate of SecYEG (*Sachelaru et al., 2013*; *Sachelaru et al., 2017*), as does the TMD of PpiD (*Miyazaki et al., 2022*), a periplasmic chaperone.

A second noteworthy advance in this work is a relatively clear view of how an $N_{cyt}$ TMD initially engages Sec61, revealing a mode of interaction very similar to SPs. In both cases, the positive-inside rule may be enforced in part by interactions with rRNA helix 59 and the anionic C-tail of eL22. The difference is that the TMD is more than twice as long as the hydrophobic region of an SP. Hence, while the N-terminal part of the TMD binds like an SP, its C-terminal part loops back through the channel pore similar to the mature domain downstream of an SP. This finding supports the through-pore model for TMD insertion rather than the sliding model, in which TMDs would instead translocate through the lipid phase at the lateral gate (*Cymer et al., 2015*). It remains to be seen whether this model also applies to $N_{exo}$ TMDs, which unlike $N_{cyt}$ TMDs, are highly refractory to inhibitors that block Sec61's lateral gate (*Smalinskaitè et al., 2022*; *O'Keefe et al., 2021*; *Zong et al., 2019*).

The third major area of insight concerns the TRAP complex and its provisionally assigned interaction with Calnexin, adding to and refining recent structural models of the TRAP complex (*Gemmer et al., 2023b*; *Jaskolowski et al., 2023*; *Pauwels et al., 2023*; *Karki et al., 2023*). We describe the extensive network of contacts with the ribosome and Sec61, potentially responsible for TRAP's effects on secretion. We show how a unique hairpin on the lumenal domain of TRAPδ forms a bridge to RPN2 in OSTA, potentially explaining why TRAP deficiency causes glycosylation defects (*Phoomak et al., 2021*). Unlike this TRAP-OSTA cooperation, we observe competition between TRAP and the BOS complex for the same binding site on Sec61 and the ribosome, explaining why TRAP is depleted and disordered in MPT-containing RTCs. Our sequence analysis and structure predictions for TRAP complexes in taxa beyond mammals reveal conserved features indicating that the original eukaryotic TRAP complex was similar to mammalian TRAPαβγ, which probably evolved from a predecessor in archaea.

These three major areas of insight, together with a number of additional findings regarding Sec61-ribosome interactions, the Sec61 pore ring, the dynamic tail of uL22, and a marked conformational change in 5.8S rRNA on translocon binding collectively lead to a wealth of new hypotheses regarding protein translocation at the ER. A major theme emerging from this and other studies in recent years

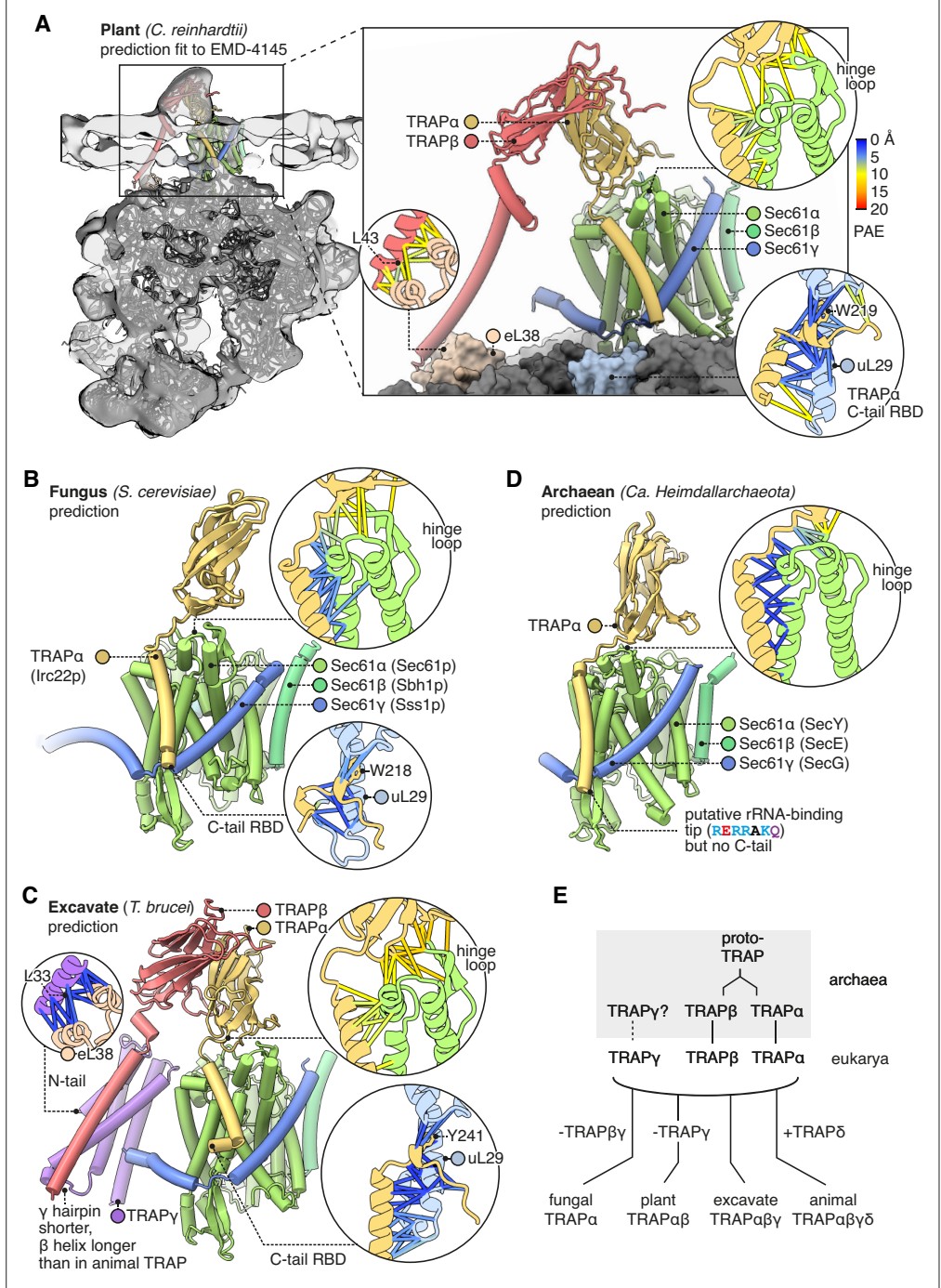

**Figure 8.** Diversity of translocon-associated protein (TRAP) structures across eukarya and archaea. (**A**) Predicted structure of a plant TRAPαβ-Sec61 complex (*C. reinhardtii*) fitted to a subtomogram average from native membranes (EMD-4145). Insets show predicted aligned errors (PAEs) for the TRAPα-Sec61α interaction and for the separately predicted TRAPα-uL29 and TRAPβ-eL38 interactions. A plant ribosome model is shown for context (PDB 8B2L). (**B**) Predicted structure of a fungal TRAPα-Sec61 complex (*S. cerevisiae*). Insets show PAEs for the TRAPα-Sec61α interaction and for the separately predicted TRAPα-uL29 interaction. (**C**) Predicted structure of an excavate TRAPαβγ-Sec61 complex (*T. brucei*). Insets show PAEs for the TRAPα-Sec61α interaction and for the separately predicted TRAPα-uL29 and TRAPγ-eL38 interactions. (**D**) Predicted structure of an archaean TRAPα-SecYEG complex (Ca. Heimdallarchaeota). The inset shows PAEs for the TRAPα-SecY interaction. (**E**) Schematic representation of the hypothesis that TRAPαβ share a common ancestor and were present in archaea, then inherited by the cenancestral eukaryote alongside TRAPγ. Typical subunit compositions are included alongside the indicated taxa, ignoring further variations that exist within each taxon.

is that the translocon is not static in either its composition or conformation. Instead, various factors and domains are often mutually exclusive, requiring dynamic reorganisation in ways small and large. Understanding how different types of substrates drive such reorganisation to facilitate their biogenesis remains a major challenge. This problem is analogous to the cytosol where multiple factors must dynamically access the ribosome exit tunnel to triage the nascent protein towards different fates (*Gamerdinger and Deuerling, 2024*). Our analysis of various new translocon configurations sheds light on how this complex machinery facilitates secretory and membrane protein biogenesis.

## Methods

### Sample preparation and electron microscopy

Sample preparation and electron microscopy was described previously (*Smalinskaitė et al., 2022*). In brief, the in vitro transcription reaction used a PCR-generated template containing the SP6 promoter (*Feng and Shao, 2018*; *Sharma et al., 2010*). The transcription reactions were for 1 hr at 37°C. The resulting transcript was used without further purification and was diluted 1:20 in the IVT reaction, which was carried out in rabbit reticulocyte lysate as described previously (*Feng and Shao, 2018*; *Sharma et al., 2010*). The reaction was supplemented with canine rough microsomes (cRMs) prepared and used as described previously (*Walter and Blobel, 1983*). The translation reaction was incubated for 30 min at 32°C, then halted by transferring the samples to ice. All further steps were performed at 0–4°C, unless stated otherwise.

A 2 ml translation reaction was divided in four, and each aliquot layered on a 500 µl cushion of 20% sucrose in 1× RNC buffer (50 mM HEPES-KOH, pH 7.5, 200 mM KOAc, 5 mM Mg(OAc)$_2$). The microsomes were sedimented by centrifugation at 4°C in the TLA-55 rotor (Beckman) at 55,000 rpm for 20 min. The cRM pellets were each resuspended in 25 µl of RNC buffer and pooled. The sample was incubated with 250 µM BMH on ice for 15 min and quenched with 5 mM 2-mercaptoethanol. The microsomes were diluted with 400 µl of solubilisation buffer (RNC buffer containing 1.5% digitonin) and incubated for 10 min on ice. The digitonin was obtained from Calbiochem and further purified as described previously (*Görlich and Rapoport, 1993*). The sample was centrifuged at 20,000×$g$ and 4°C for 15 min. The supernatant was transferred to a tube containing 20 µl of StrepTactin High Performance Sepharose beads (GE Healthcare) and incubated for 1.5 hr at 4°C. The resin was then washed five times with 0.5 ml RNC buffer containing 0.25% digitonin and eluted by incubation for 1 hr on ice with 40 µl of RNC buffer containing 0.25% digitonin and 50 mM biotin. The absorbance of the eluate for both samples was 3.4 at 260 nm.

The affinity-purified RNCs were vitrified on UltrAuFoil R 1.2/1.3 300-mesh grids (Quantifoil) coated with graphene oxide (Sigma-Aldrich). In a Vitrobot Mark IV (Thermo Fisher Scientific) at 4°C and 100% ambient humidity, each grid was loaded with 3 µl of sample, blotted 4 s with Whatman filter papers at a blot force of –15, and plunge-frozen in liquid ethane at 92 K. Automated data collection was performed on a Titan Krios microscope (Thermo Fisher Scientific) equipped with an XFEG source operating at an accelerating voltage of 300 kV. Defocus was programmed to range between 2.7 and 1.9 µm. Movies were captured using a K3 Bioquantum direct electron detector (Gatan) operating in super-resolution mode. Movies were dose-fractionated into 54 frames covering a total dose of 54 e⁻ Å⁻². 17,540 images were collected.

### Image processing

Movie frames were motion-corrected using MotionCor2 (*Zheng et al., 2017*) with 7×5 patches and dose-weighting, and their contrast transfer functions (CTFs) were fit using CTFFIND 4.1 (*Rohou and Grigorieff, 2015*). After manual curation, 285 of 12,540 micrographs (2.3%) were discarded due to poor CTF fits or thick ice. Subsequent steps were performed in RELION-4.0 (*Kimanius et al., 2021*). Manual picking on 20 randomly selected micrographs yielded 2745 particles, which were used to train the automatic particle-picker Topaz (*Bepler et al., 2019*). Topaz was run on all micrographs and picks assigned a figure of merit (FOM) above –3 were retained. This cutoff was chosen based on a histogram of pick FOMs, which was approximately normal above –3 but displayed a long lower-FOM tail, and subsequently checked against the micrographs to verify sensible results. Picked particles were extracted in 412 px boxes (1.34 Å px⁻¹), binned to 128 px (4.31 Å px⁻¹), and classified in 2D with a 300 Å diameter mask, 200 classes $k$ and a regularisation parameter $T$ of 2. Eighty 2D classes showing

clear molecular features were retained, encompassing 1,188,459 particles. These particles' coordinates were used to retrain Topaz. The retrained Topaz model picked 1,389,410 particles at FOM ≥ –1, a cutoff selected by the same criterion as above.

The picked particles were extracted in 412 px boxes, binned to 128 px, and refined in 3D against a mammalian ribosome reference map lowpass-filtered at 70 Å, yielding a Nyquist-limited (8.62 Å) map. With fixed alignments from this refinement, particles were classified in 3D (*k*=20, *T*=4), yielding 80S (38%), 80S ratcheted (15%), 60S (10%), poorly resolved (36%), and 10 noise (1%) classes, as well as 6 empty classes. The 875,065 well-resolved ribosomal particles were combined, re-extracted in 420 px boxes without binning, and refined in 3D to obtain a 2.86 Å map, which improved to 2.68 Å after CTF refinement and Bayesian polishing. This is the all-particle map.

The aligned particles were then subclassified without realignment using three different masks for residual signal subtraction: a tight mask surrounding the mobile parts of Sec61 and RAMP4, a tight mask surrounding the TRAP complex, and a loose mask encompassing both Sec61•RAMP4 and TRAP. The loose masking did not clearly separate most classes, but did provide the clearest view of TRAPα's TMD, which crosses the boundary between the tighter masks. The resulting classes are shown in the processing flowchart (*Figure 1—figure supplement 1C*). The well-resolved TRAP classes were further subclassified to obtain maps where Sec61 was also more homogeneous. This second subclassification yielded noisier classes because the particle numbers are much smaller, but the TRAP and Rho-bound Sec61 classes yielded clear density. After reconstruction, density maps' local scale and occupancy were estimated using OccuPy (*Forsberg et al., 2023*). Maps were postprocessed using Relion's MTF-correction and automated B-factor sharpening, and were also lowpass-filtered where indicated. Some maps were in parallel postprocessed using DeepEMhancer (*Sanchez-Garcia et al., 2021*) for use in rendering figures.

## Molecular modelling

The 60S subunit and P-site tRNA from PDB 7TM3 were used as an initial model for the rabbit ribosome. An initial model for uL22 was fetched from the AlphaFold DB (*Tunyasuvunakool et al., 2021*). For the Sec61-RAMP4 complex and TRAP complex, initial models were generated using ColabFold2 (*Mirdita et al., 2021*) with AlphaFold2-Multimer v3 (*Evans et al., 2021*). Default options were used, except that the top-scoring models were refined by AMBER relaxation to avoid clashes that would interfere with subsequent simulations. Separate multimer predictions were also generated for the interactions of TRAP with Sec61α, TRAPγ with eL38, and TRAPα with uL23/29, for use as references for modelling restraints. All sequences were fetched from UniProt (*UniProtConsortium T, 2018*).

The initial models for these complexes were fit to density using ISOLDE (*Croll, 2018*) molecular dynamics flexible fitting with adaptive predicted local distance difference test (pLDDT)- and PAE-dependent restraints (*Croll and Read, 2021*). Segments that were not confidently predicted, as indicated by high PAE or low pLDDT scores, were omitted unless they could be built based on the cryo-EM density alone. Final real-space refinements were performed using PHENIX (*Liebschner et al., 2019*). Three rounds of global minimisation and group B-factor refinement were performed, with tight secondary structure, reference model, rotamer, and Ramachandran restraints applied. Secondary structure and reference model restraints were determined from the starting models. Hydrogen-bonding and base-pair and stacking parallelity restraints were applied to the rRNA. Final model statistics are provided in *Supplementary file 1*. Models were rendered using ChimeraX (*Pettersen et al., 2021*).

No refinements were performed for the BOS complex model, nor the four non-mammalian TRAP•Sec61 models. The *Cr* TRAP model predicted by AF2 was fitted to the tomographic density map EMD-4145 in ISOLDE using its ribosome interactions as anchor points (the Sec61-ribosome binding site and predicted TRAPβ-eL38 binding site). For each non-mammalian TRAP•Sec61 model, separate predictions were run to probe for interactions with eL38 or uL29. For the panel of multimeric predictions used to test whether RAMP4's predicted interaction with Sec61 was a positive outlier, a separate prediction was run for Sec61 with each of the other sequences in the panel. The resulting predictions were checked visually to confirm that none besides RAMP4 engaged the lateral gate, and a Python script was used to gather the intersubunit PAEs and generate a violin plot of their distribution.

## Cell culture

Flp-In T-Rex 293 cells (Invitrogen) were maintained in DMEM supplemented with 10% FBS (Gemini Foundation), and 10,000 U ml$^{-1}$ penicillin and 10 mg ml$^{-1}$ streptomycin mixture (Invitrogen and Gemini). Cells were checked approximately every 6 months for mycoplasma contamination using the Universal Mycoplasma Detection Kit (ATCC) and verified to be negative.

## Cell lines

TMCO1 knockout Flp-In T-Rex 293 cells were generated by transfection of plasmid pX330-U6-Chimeric_BB-CBh-hSpCas9 (PX330) (Addgene) encoding the guide RNA 5'-GAAACAATAACAGAGT CAGC-3'. 48 hr after transfection, cells were single-cell sorted into 96-well plates for clonal isolation. After expansion, clones were screened for successful knockout by western blotting and genomic sequencing. RAMP4 knockout Flp-In T-Rex 293 cells were generated by transfection of plasmid pSPCas9(BB)-2A-Puro (PX459) (Addgene) encoding the guide RNA 5'-AGCAAAGGATCCGTAT GGCC-3'. 24 hr after transfection, cells were selected in puromycin (1 µg ml$^{-1}$) for 72 hr followed by single-cell sorting into 96-well plates for clonal isolation. Clones were verified as above. Stable Flp-In T-Rex 293 cells containing a doxycycline-inducible 3xFlag-TMCO1 (NP_061899.3) or 3xFlag-RAMP4 (NP_055260.1) construct were generated using the Flp-In system (Invitrogen) according to the manufacturer's instructions. Briefly, pOG44 and the appropriate pcDNA5/FRT/TO-based construct were co-transfected into the respective knockout cells. 24 hr after transfection, cells were selected for successful Flp-mediated recombination with 100 µg ml$^{-1}$ hygromycin B for 2 weeks. After expansion, the pooled population of resistant cells was assessed for inducible expression at near-native levels. For immunoprecipitation, cells were induced with doxycycline at 1 ng ml$^{-1}$ for 48 hr (3xFlag-TMCO1) or 1 ng ml$^{-1}$ for 24 hr (3xFlag-RAMP4).

## Interaction analysis

Microsomes from wild-type cells and cells expressing 3xFlag-tagged TMCO1 or RAMP4 were prepared as previously described (*Sundaram et al., 2022*) except that the micrococcal nuclease digestion was performed with 10,000 U of micrococcal nuclease (NEB), 3 U DNAse (Promega), 1 mM CaCl$_2$, and 0.6 mM PMSF, and incubated at room temperature for 20 min before quenching with 2.5 mM EGTA. Microsomes (750 µl at $A_{260}$ = 50) were solubilised in insertion buffer (50 mM HEPES-KOH pH 7.5, 10 mM MgCl$_2$, 250 mM KOAc, 250 mM sucrose) supplemented with 2.5% digitonin and 1× protease inhibitor cocktail (Roche, 11836170001) for 45 min on ice and then diluted twice with insertion buffer containing 150 mM KOAc. Digitonin-solubilised microsomes were cleared by centrifugation at 12,500×*g* for 15 min at 4°C. The cleared supernatant was immunoprecipitated in batch format using 75 µl M2 Flag affinity beads (Sigma, A2220) and gentle agitation overnight at 4°C. The unbound fraction was removed and beads were washed three times with 600 µl of insertion buffer supplemented with 0.4% digitonin. Bound material was eluted twice, for 45 min on ice, with 150 µl of insertion buffer containing 200 mM KOAc supplemented with 0.5 mg ml$^{-1}$ Flag peptide (ApexBio, A6001) and 0.4% digitonin. The eluate was collected using a pre-equilibrated spin filter column (Thermo Fisher, 69725). The ribosome-bound fraction was obtained by pelleting the eluate through a 300 µl sucrose cushion (50 mM HEPES pH 7.4, 10 mM MgCl$_2$, 150 mM KCl, 500 mM sucrose, and 0.4% digitonin) at 355,000×*g* for 1 hr at 4°C in a TLA120.1 rotor. The affinity-purified ribosome fraction was analysed by immunoblotting together with 1% of the starting microsomes using antibodies against the following antigens at the indicated dilutions: RAMP4 (Abcam #ab184571, which recognizes both RAMP4 homologs; 1:5000), uL22 (Abgent, AP9892b; 1:1000), Sec61β (*Fons et al., 2003*; 1:10,000), TRAPα (*Fons et al., 2003*; 1:5000), STT3A (Novus Biologicals, H00003703-M02; 1:1000), TMCO1 (*Anghel et al., 2017*; 1:5000), CCDC47 (Bethyl Laboratories, A305-100A; 1:2000), and NOMO (Invitrogen; PA5-47534; 1:1000).

## Materials availability

Materials introduced in this study are available from the corresponding author upon request.

## Acknowledgements

We are grateful to Min Kyung Kim and the staff of LMB's EM facility for collecting the dataset on which this study's analysis is based; J Grimmett and T Darling of LMB's Scientific Computing for support. Funding: This work was supported by the Medical Research Council as part of United Kingdom Research and Innovation (MC_UP_A022_1007 to RSH) and NIH R35 GM145374 to RJK FZ was supported by NIH training grant T32 GM007183.

## Additional information

### Competing interests

Ramanujan S Hegde: Scientific advisor and equity holder of Gate Bioscience. The other authors declare that no competing interests exist.

### Funding

| Funder | Grant reference number | Author |
| --- | --- | --- |
| Medical Research Council | MC_UP_A022_1007 | Ramanujan S Hegde |
| National Institutes of Health | R35 GM145374 | Robert J Keenan |
| National Institutes of Health | T32 GM007183 | Frank Zhong |

The funders had no role in study design, data collection and interpretation, or the decision to submit the work for publication.

### Author contributions

Aaron JO Lewis, Conceptualization, Investigation, Visualization, Methodology, Writing – original draft, Writing – review and editing; Frank Zhong, Investigation, Methodology; Robert J Keenan, Supervision, Funding acquisition, Writing – review and editing; Ramanujan S Hegde, Conceptualization, Supervision, Project administration, Writing – review and editing

### Author ORCIDs

Aaron JO Lewis (ID) https://orcid.org/0000-0001-8818-1763
Robert J Keenan (ID) http://orcid.org/0000-0003-1466-0889
Ramanujan S Hegde (ID) https://orcid.org/0000-0001-8338-852X

Reviewer #1 (Public review): https://doi.org/10.7554/eLife.95814.3.sa1
Reviewer #2 (Public review): https://doi.org/10.7554/eLife.95814.3.sa2
Author response https://doi.org/10.7554/eLife.95814.3.sa3

## Additional files

### Supplementary files

- Supplementary file 1. Data collection, refinement, model, and validation statistics.
- MDAR checklist

### Data availability

The EM maps have been deposited in EMDB with the accession numbers EMD-19195, EMD-19196, EMD-19197, EMD-19198, 19199, EMD-19200, EMD-19201, EMD-19202, EMD-19203, and EMD-19204. The molecular models have been deposited in the PDB with accession numbers 8RJB, 8RJC, and 8RJD. The AlphaFold2-predicted structures (*Varadi et al., 2024*) reported in this study have been deposited in ModelArchive with accession numbers ma-jjnuw, ma-hbsof, ma-hknzh, ma-x3uvj, ma-j8wag, ma-9gsmk, ma-l9qqq, and ma-ise4t.

The following datasets were generated:

| Author(s) | Year | Dataset title | Dataset URL | Database and Identifier |
|---|---|---|---|---|
| Lewis AJO, Hegde RS | 2024 | Structure of the rabbit 80S ribosome stalled on a 2-TMD rhodopsin intermediate in complex with Sec61-RAMP4 | https://www.ebi.ac.uk/emdb/EMD-19195 | Electron Microscopy Data Bank, EMD-19195 |
| Lewis AJO, Hegde RS | 2024 | Structure of the rabbit 80S ribosome stalled on a 2-TMD rhodopsin intermediate in complex with Sec61, all-particle reconstruction | https://www.ebi.ac.uk/emdb/EMD-19196 | Electron Microscopy Data Bank, EMD-19196 |
| Lewis AJO, Hegde RS | 2024 | Structure of the rabbit 80S ribosome stalled on a 2-TMD rhodopsin intermediate in complex with Sec61-TRAP, open conformation 1 | https://www.ebi.ac.uk/emdb/EMD-19197 | Electron Microscopy Data Bank, EMD-19197 |
| Lewis AJO, Hegde RS | 2024 | Structure of the rabbit 80S ribosome stalled on a 2-TMD rhodopsin intermediate in complex with Sec61-TRAP, open conformation 2 | https://www.ebi.ac.uk/emdb/EMD-19198 | Electron Microscopy Data Bank, EMD-19198 |
| Lewis AJO, Hegde RS | 2024 | Structure of the rabbit 80S ribosome stalled on a 2-TMD rhodopsin intermediate in complex with Sec61, bound TRAP-alpha TMD | https://www.ebi.ac.uk/emdb/EMD-19199 | Electron Microscopy Data Bank, EMD-19199 |
| Lewis AJO, Hegde RS | 2024 | Structure of the rabbit 80S ribosome stalled on a 2-TMD rhodopsin intermediate in complex with Sec61-TRAP, conformation 1 | https://www.ebi.ac.uk/emdb/EMD-19200 | Electron Microscopy Data Bank, EMD-19200 |
| Lewis AJO, Hegde RS | 2024 | Structure of the rabbit 80S ribosome stalled on a 2-TMD rhodopsin intermediate in complex with Sec61-TRAP, conformation 2 | https://www.ebi.ac.uk/emdb/EMD-19201 | Electron Microscopy Data Bank, EMD-19201 |
| Lewis AJO, Hegde RS | 2024 | Structure of the rabbit 80S ribosome stalled on a 2-TMD rhodopsin intermediate in complex with Sec61, closed conformation 1 | https://www.ebi.ac.uk/emdb/EMD-19202 | Electron Microscopy Data Bank, EMD-19202 |
| Lewis AJO, Hegde RS | 2024 | Structure of the rabbit 80S ribosome stalled on a 2-TMD rhodopsin intermediate in complex with Sec61, closed conformation 2 | https://www.ebi.ac.uk/emdb/EMD-19203 | Electron Microscopy Data Bank, EMD-19203 |
| Lewis AJO, Hegde RS | 2024 | Structure of the rabbit 80S ribosome stalled on a 2-TMD rhodopsin intermediate in complex with Rho-bound Sec61 | https://www.ebi.ac.uk/emdb/EMD-19204 | Electron Microscopy Data Bank, EMD-19204 |

*Continued on next page*

*Continued*

| Author(s) | Year | Dataset title | Dataset URL | Database and Identifier |
|---|---|---|---|---|
| Lewis AJO, Hegde RS | 2024 | Structure of the rabbit 80S ribosome stalled on a 2-TMD rhodopsin intermediate in complex with Sec61-RAMP4 | https://www.rcsb.org/structure/8RJB | RCSB Protein Data Bank, 8RJB |
| Lewis AJO, Hegde RS | 2024 | Structure of the rabbit 80S ribosome stalled on a 2-TMD rhodopsin intermediate in complex with Sec61-TRAP, open conformation 1 | https://www.rcsb.org/structure/8RJC | RCSB Protein Data Bank, 8RJC |
| Lewis AJO, Hegde RS | 2024 | Structure of the rabbit 80S ribosome stalled on a 2-TMD rhodopsin intermediate in complex with Sec61-TRAP, open conformation 2 | https://www.rcsb.org/structure/8RJD | RCSB Protein Data Bank, 8RJD |
| Lewis AJO, Hegde RS | 2024 | *Chlamydomonas reinhardtii* Sec61-RAMP4 complex | https://modelarchive.org/doi/10.5452/ma-jjnuw | ModelArchive, ma-jjnuw |
| Lewis AJO, Hedge RS | 2024 | *Saccharomyces cerevisiae* Sec61-RAMP4 complex (aka Ysy6p) | https://modelarchive.org/doi/10.5452/ma-hbsof | ModelArchive, ma-hbsof |
| Lewis AJO, Hegde RS | 2024 | *Canis lupus* familiaris Sec61-RAMP4 complex | https://modelarchive.org/doi/10.5452/ma-hknzh | ModelArchive, ma-hknzh |
| Lewis AJO, Hegde RS | 2024 | *Heimdallarchaean* TRAPα-SecY complex | https://modelarchive.org/doi/10.5452/ma-x3uvj | ModelArchive, ma-x3uvj |
| Lewis AJO, Hegde RS | 2024 | *Trypanosoma brucei* TRAP-Sec61 complex | https://modelarchive.org/doi/10.5452/ma-j8wag | ModelArchive, ma-j8wag |
| Lewis AJO, Hegde RS | 2024 | *Chlamydomonas reinhardtii* TRAP-Sec61-RAMP4-eL38 complex | https://modelarchive.org/doi/10.5452/ma-9gsmk | ModelArchive, ma-9gsmk |
| Lewis AJO, Hegde RS | 2024 | *Saccharomyces cerevisiae* Sec61-TRAP complex (aka Irc22p) | https://modelarchive.org/doi/10.5452/ma-l9qqq | ModelArchive, ma-l9qqq |
| Lewis AJO, Hegde RS | 2024 | *Canis lupus familiaris* BOS complex (TMEM147/Nicalin/NOMO, C. lupus familiaris) | https://modelarchive.org/doi/10.5452/ma-ise4t | ModelArchive, ma-ise4t |

The following previously published datasets were used:

| Author(s) | Year | Dataset title | Dataset URL | Database and Identifier |
|---|---|---|---|---|
| Tesina P, Ebine S, Buschauer R, Thoms M, Matsuo Y, Inada T, Beckmann R | 2023 | Molecular basis of eIF5A-dependent CAT tailing in eukaryotic ribosome-associated quality control | https://www.rcsb.org/structure/8AGX | RCSB Protein Data Bank, 8AGX |
| Smirnova J, Loerke J, Kleinau G, Schmidt A, Bürger J, Meyer EH, Mielke T, Scheerer P, Bock R, Spahn CMT, Zoschke R | 2023 | Structure of the actively translating plant 80S ribosome at 2.2 Å resolution | https://www.rcsb.org/structure/8B2L | RCSB Protein Data Bank, 8B2L |

*Continued on next page*

*Continued*

| Author(s) | Year | Dataset title | Dataset URL | Database and Identifier |
|---|---|---|---|---|
| Braunger K, Pfeffer S, Shrimal S, Gilmore R, Berninghausen O, Mandon EC, Becker T, Förster F, Beckmann R | 2018 | Structural basis for coupling protein transport and N-glycosylation at the mammalian endoplasmic reticulum | https://www.rcsb.org/structure/6FTI | RCSB Protein Data Bank, 6FTI |
| Holm M, Natchiar SK, Rundlet EJ, Myasnikov AG, Watson ZL, Altman RB, Wang HY, Taunton J, Blanchard SC | 2023 | mRNA decoding in human is kinetically and structurally distinct from bacteria | https://www.rcsb.org/structure/8GLP | RCSB Protein Data Bank, 8GLP |
| Holm M, Natchiar SK, Rundlet EJ, Myasnikov AG, Watson ZL, Altman RB, Wang HY, Taunton J, Blanchard SC | 2023 | mRNA decoding in human is kinetically and structurally distinct from bacteria | https://www.ebi.ac.uk/emdb/EMD-40205 | EMDB, EMD-40205 |
| Pfeffer S, Dudek J, Schaffer M, Ng BG, Albert S, Plitzko JM, Baumeister W, Zimmermann R, Freeze HH, Engel BD, Forster F | 2017 | Dissecting the molecular organization of the translocon-associated protein complex | https://www.ebi.ac.uk/emdb/EMD-4145 | EMDB, EMD-4145 |
| Braunger K, Pfeffer S, Shrimal S, Gilmore R, Berninghausen O, Mandon EC, Becker T, Forster F, Beckmann R | 2018 | Structural basis for coupling protein transport and N-glycosylation at the mammalian endoplasmic reticulum | https://www.ebi.ac.uk/emdb/EMD-4311 | EMDB, EMD-4311 |
| Voorhees RM, Hegde RS | 2016 | The structure of the mammalian Sec61 channel opened by a signal sequence | https://www.rcsb.org/structure/3JC2 | RCSB Protein Data Bank, 3JC2 |
| Voorhees RM, Hegde RS | 2016 | The structure of the mammalian Sec61 channel opened by a signal sequence | https://www.ebi.ac.uk/emdb/EMD-3245 | EMDB, EMD-3245 |
| Smalinskaitė L, Kim MK, Lewis AJO, Keenan RJ, Hegde RS | 2022 | Mechanism of an intramembrane chaperone for multipass membrane proteins | https://www.rcsb.org/structure/7TM3 | RCSB Protein Data Bank, 7TM3 |
| Smalinskaite L, Kim MK, Lewis AJO, Keenan RJ, Hegde RS | 2022 | Mechanism of an intramembrane chaperone for multipass membrane proteins | https://www.ebi.ac.uk/emdb/EMD-25994 | EMDB, EMD-25994 |
| Gemmer M, Chaillet ML, van Loenhout J | 2023 | Visualization of translation and protein biogenesis at the ER membrane | https://www.ebi.ac.uk/emdb/EMD-15885 | EMDB, EMD-15885 |
| Gemmer M, Chaillet ML, van Loenhout J | 2023 | Visualization of translation and protein biogenesis at the ER membrane | https://www.ebi.ac.uk/emdb/EMD-15889 | EMDB, EMD-15889 |
| Gemmer M, Fedry JMM, Forster FG | 2022 | Subtomogram average of the human Sec61-TRAP-OSTA-translocon | https://www.rcsb.org/structure/8B6L | RCSB Protein Data Bank, 8B6L |

| Author(s) | Year | Dataset title | Dataset URL | Database and Identifier |
|---|---|---|---|---|
| Gemmer M, Fedry JMM, Forster FG | 2022 | Subtomogram average of the human Sec61-TRAP-OSTA-translocon | https://www.ebi.ac.uk/emdb/EMD-15870 | EMDB, EMD-15870 |

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
