## [Editor Report · eLife assessment]

This **landmark** work by Lewis et al. represents the most significant breakthrough in membrane and secretory biogenesis in recent years. Their work reveals with outstanding clarity how nascent transmembrane segments can pass through the gate of Sec61 into the ER membrane through the coordinated motions of a conformationally and compositionally dynamic machine. Among many other insights, the authors discovered how a new factor, RAMP4, contributes to the formation and function of the lateral gate for certain substrates. The technical quality of the work is **exceptional**, setting the bar appropriately high.

---

## [Referee Report · Reviewer #1 (Public review)]

The paper meticulously explores various conformations and states of the ribosome-translocon complex. Employing advanced techniques such as cryoEM structural determination and AlphaFold modeling, the study delves into the dynamic nature of the ribosome-translocon complex. The findings from these analyses unveil crucial insights, significantly advancing our understanding of the co-translational translocation process in cellular mechanisms.

To begin with, the authors employed a construct comprising the first two transmembrane domains of rhodopsin as a model for studying protein translocation. They conducted in vitro translation, followed by the purification of the ribosome-translocon complex, and determined its cryoEM structures. An in-depth analysis of their ribosome-translocon complex structure revealed that the nascent chain can pass through the lateral gate of translocon Sec61, akin to the behavior of a Signaling Peptide. Additionally, Sec61 was found to interact with 28S rRNA helix 24 and the ribosomal protein uL24. In summary, their structural model aligns with the through-pore model of insertion, contradicting the sliding model.

Secondly, the authors successfully identified RAMP4 in their ribosome-translocon complex structure. Notably, the transmembrane domain of RAMP4 mimics the binding of a Signaling Peptide at the lateral gate of Sec61, albeit without unplugging. Intriguingly, RAMP4 is exclusively present in the non-multipass translocon ribosome-translocon complex, not in those containing multipass translocon. This observation suggests that co-translational translocation specifically occurs in the Sec61 channel that includes bound RAMP4. Additionally, the authors discovered an interaction between the C-tail of ribosomal proteins uL22 and the translocon Sec61, providing valuable insights into the nascent chain's behavior.

Moving on to the third point, the focused classification unveiled TRAP complex interactions with various components. The authors propose that the extra density observed in their novel ribosome-translocon complex can be attributed to calnexin, a major binder of TRAP according to previous studies. Furthermore, the new structure reveals a TRAP-OSTA interaction. This newly identified TRAP-OSTA interaction offers a potential explanation for why patients with TRAP delta defects exhibit congenital disorders of glycosylation.

In conclusion, this paper presents a robust contribution to the field with its thorough structural and modeling analyses. The significance of the findings is evident, providing valuable insights into the intricate mechanisms of protein co-translational translocation. The well-crafted writing, meticulous analyses, and clear figures collectively contribute to the overall strength of the paper.

---

## [Referee Report · Reviewer #2 (Public review)]

Summary:

In the manuscript Lewis and Hegde present a structural study of the ribosome-bound multipass translocon (MPT) based on re-analysis of cryo-EM single particle data of ribosome-MPTs processing the multipass transmembrane substrate RhoTM2 from a previous publication (Smalinskaité et al, Nature 2022) and AlphaFold2 multimer modeling. Detailed analysis of the laterally open Sec61 is obtained from PAT-less particles.

The following major claims are made:

- TMs can bind similarly to the Sec61 lateral gate as signal peptides.

- Ribosomal H59 is in immediate proximity to basic residues of TMs and signal peptides, suggesting it may contribute to the positive-inside rule.

- RAMP4/SERP1 binds to the Sec61 lateral gate and the ribosome near 28S rRNA's helices 47, 57, and 59 as well as eL19, eL22, and eL31.

- uL22 C-terminal tail binds H24/47 blocking a potential escape route for nascent peptides to the cytosol.

- TRAP and BOS compete for binding to Sec61 hinge.

- Calnexin TM binds to TRAPg.

- NOMO wedges between TRAP and MPT.

Strengths:

The manuscript contains numerous novel new structural analyses and their potential functional implications. While all findings are exciting, the highlight is the discovery of RAMP4/SERP1 near the Sec61 lateral gate. Overall, the strength is the thorough and extensive structural analysis of the different high-resolution RTC classes as well as the expert bioinformatic evolutionary analysis.

---

## [Author Response]

The following is the authors’ response to the original reviews.

**Reviewer #1 (public review and recommendations for the authors):**
Major points:(1) The identification of RAMP4 is a pivotal discovery in this paper. The sophisticated AlphaFold prediction, de novo model building of RAMP4's RBD domain, and sequence analyses provide strong evidence supporting the inclusion of RAMP4 in the ribosome-translocon complex structure.

However, it is crucial to ensure the presence of RAMP4 in the purified sample. Particularly, a validation step such as western blotting for RAMP4 in the purified samples would strengthen the assertion that the ribosome-translocon complex indeed contains RAMP4. This is especially important given the purification steps involving stringent membrane solubilization and affinity column pull-down.

As suggested, we have added Western blots showing that RAMP4 is retained at secretory translocons (and not multipass translocons) after solubilisation, affinity purification, and recovery of ribosome-translocon complexes (Fig. 3F). This data supports both our assignment of RAMP4 in ribosome-translocon complexes, and also the structure-based proposition that its occupancy is mutually exclusive with the multipass translocon (in particular, the PAT complex).

(2) Despite the comprehensive analyses conducted by the authors, it is challenging to accept the assertion that the extra density observed in TRAP class 1 corresponds to calnexin. The additional density in TRAP class 1 appears to be less well-resolved, and the evidence for assigning it as calnexin is insufficient. The extra density there can be any proteins that bind to TRAP. It is recommended that the authors examine the density on the ER lumen side. An investigation into whether calnexin's N-globular domain and P-domain are present in the ER lumen in TRAP class 1 would provide a clearer understanding.

We agree that the Calnexin assignment is less confident than the other assignments in this manuscript, and that further support would be ideal. We have exhaustively searched our maps for any unexplained density connected with the putative Calnexin TMD, and have found none. This is consistent with Calnexin's lumenal domain being flexibly linked to its TMD, and thus would not be resolved in a ribosome-aligned reconstruction.

Our assignment of this TMD to Calnexin was based on existing biochemical data (referenced in the paper) favouring this as the best working hypothesis by far: Calnexin is TRAP’s only abundant co-purifying factor, and their interaction is sensitive to point mutations in the Calnexin TMD. Recognising that this is not conclusive, we have ensured that the text and figures consistently describe this assignment as provisional or putative.

(3) In the section titled 'TRAP competes and cooperates with different translocon subunits,' the authors present a compelling explanation for why TRAP delta defects can lead to congenital disorders of glycosylation. To enhance this explanation, it would be valuable if the authors could provide additional analyses based on mutations mentioned in the references. Specifically, examining whether these mutations align with the TRAP delta-OSTA structure models would strengthen the link between TRAP delta defects and the observed congenital disorders of glycosylation.

We agree that mapping disease-causing point mutants to the TRAP delta structure could be potentially informative. Unfortunately, the referenced TRAP delta disease mutants act by simply impairing TRAP delta expression, and thus admit no such fine-grained analyses. However, sequence conservation is our next best guide to mutant function. We note in the text that the contact site charges on TRAP delta and RPN2 are conserved, and that the closest-juxtaposed interaction pair (K117 on TRAPδ and D386 on RPN2) is also the most conserved.

Here are some minor points:(1) In the introduction, when the EMC, PAT, and BOS complexes were initially mentioned, it would be beneficial for the authors to provide more context or cite relevant references. This additional information will aid readers in better understanding these complexes, ensuring a smoother comprehension of their significance in the context of the study.

The Introduction has been edited to provide more context with relevant references.

(2) In Figure 7, it would be valuable for the authors to include details on how they sampled the sequence alignments.

To clarify this methodological point, we have revised the Figure 7 caption to include these sentences: “The logo plots in panels A and D represent an HMM generated by jackHMMER upon convergence after querying UniProtKB’s metazoan sequences with the human TRAPα sequence. Only signal above background is shown, as rendered by Skylign.org.”

**Reviewer #2 (public review and recommendations for the authors):**
Strengths:The manuscript contains numerous novel new structural analyses and their potential functional implications. While all findings are exciting, the highlight is the discovery of RAMP4/SERP1 near the Sec61 lateral gate. Overall, the strength is the thorough and extensive structural analysis of the different high-resolution RTC classes as well as the expert bioinformatic evolutionary analysis.Weaknesses:A minor downside of the manuscript is the sheer volume of analyses and mechanistic hypotheses, which makes it sometimes difficult to follow. The authors might consider offloading some analyses based on weaker evidence to the supplement to maximize impact.

We agree that the manuscript is long, but we have retained what we feel are the most important findings in the main text because the supplement is often undiscoverable via literature searches. Indeed, we chose eLife for its flexibility regarding article length and suitability for extended and detailed analyses.

Major:- Figure S1 does not capture the fact that a PAT-free subset of particles is analyzed. The PAT classification step should be added.

We apologise for having caused some confusion on this point: we do not show a PAT classification step because there was none. Instead we reanalysed the whole dataset with a focus on Sec61 and TRAP. The very little PAT present (9% of particles, per Smalinskaitė et al. 2022) appeared as a very weak density in some of the closed-Sec and weak-TRAP classes.

- The assignment of calnexin appears highly speculative. As the authors acknowledge the EM density is clearly of insufficient resolution for identification, and also AF2 does not render orthogonal support for the interpretation. The binding to TRAPg also does not explain complex formation in lower eukaryotes that do not have TRAPg. The authors may consider moving the calnexin assignment and interpretation to the supplement as it appears highly speculative. In any case, it should not be referred to as a hypothesis and not a structure.

We agree that the Calnexin assignment is less confident than the other assignments in this manuscript, and that further support would be ideal. Our assignment of this TMD to Calnexin was based on existing biochemical data (referenced in the paper) favouring this as the best working hypothesis by far: Calnexin is TRAP’s only abundant co-purifying factor, and their interaction is sensitive to point mutations in the Calnexin TMD. Recognising that this is not conclusive, we have ensured that the text and figures consistently describe this assignment as provisional or putative.

- P. 8: "This extensive competition explains why prior studies found TRAP in only 40% of MPT complexes, but at high occupancy at all other RTCs29". The interpretation is at odds with a recent re-analysis of the same dataset (preprint: Gemmer et al 2023, https://doi.org/10.1101/2023.11.28.569136), which finds TRAP occupancy to negatively correlate with PAT, not BOS.

The reviewer is correct that the Gemmer study demonstrates a negative correlation between PAT and TRAP occupancy, but it does not, as the reviewer claims, argue against a negative correlation between BOS and TRAP. In fact it agrees that Sec61•BOS•PAT complex would clash with TRAP, and that therefore “BOS could trigger release of TRAP from the multipass translocon.” Thus, there is no conflict between the two studies. The revised text in this passage now cites the Gemmer et al. preprint and clarifies that TRAP is partially displaced by competition with BOS, but retained at the translocon via its ribosome-binding domain.

- P. 7/8: the authors suggest that TRAPd may be important for OSTA recruitment and hence TRAPd deletion may cause glycosylation defects in patients by failure to recruit OSTA. However, cryo-ET studies (Pfeffer et al, Nat. Comms 2017) showed that OSTA still binds in patient-derived microsomes (and the OSTA-TRAPd interaction). The author should discuss their model in the light of these data.

As explained in the text, our hypothesis predicts that TRAPδ is more important for OSTA’s recruitment to the RTC than for its RTC affinity: “OSTA’s attraction to TRAPδ is weak compared to its binding to the ribosome, but TRAPδ may nonetheless help recruit OSTA, since TRAPδ would attract OSTA from most possible angles of approach, whereas OSTA’s ribosome contacts are stereospecific.” Therefore the fact that Pfeffer et al. 2017 found OSTA at some TRAPδ-negative RTCs is not surprising. For confirmation we would look for TRAPδ-dependent glycosylation sites in fast-folding domains or otherwise kinetically sensitive loci, and indeed TRAP-dependence screens return complex profiles that could be consistent with such a mechanism (Phoomak et al. 2021).

- Some confidence measure for the assignment of SERP1/RAMP4 should be provided adding support for the claim "The resolution of the RBD density was sufficient for de novo modelling". Indeed, the N-terminal ribosome-bound segment appears well resolved and programs like Modelangelo or FindMySequence should provide a confidence measure for the assignment of the density to SERP1. The TM part appears less well resolved, but the connectivity to the Nterminus may justify the assignment, which should be elaborated on.

Although we appreciate the value of tools like Modelangelo or FindMySequence, and would have used them if we were resting our assignment of RAMP4 on its RBD alone, we feel that such analyses would be superfluous here. They would quantify only the buildability of RAMP4’s

RBD, whereas the real question of RAMP4’s assignability is independently supported by AlphaFold’s confirmation of RAMP4’s TMD as the Sec61-binding density, and further biochemical data provided or cited in the paper.

- P. 3: "Because PAT complex recruitment and MPT assembly are just beginning, ..." the implicit kinetic model seems to be that the MPT subcomplexes assemble on ribosome and Sec61. What is the evidence for this model and later recruitment of PAT (as opposed to GEL, BOS, and PAT binding pre-assembled)?

The work of Sundaram et al. (PMID 36261522) established that PAT, GEL and BOS do not coassociate appreciably in the absence of the ribosome-Sec61 complex. This is consistent with the structural data in Smalinskaite et al. (PMID 36261528), which shows that PAT, GEL, and BOS each contact the ribosome (and Sec61 in the case of PAT and BOS), but have few if any specific contacts among themselves. Finally, data in both of these studies show that recruitment of each complex to the RNC is not lost when any of them is missing, arguing that each is capable of independent recruitment to ribosome-Sec61 complexes.

- p. 4: the meaning of the sentence "Stabilising interactions with this widely conserved motif may help Sec61 respond to its diverse substrates with a consistent open state." is not entirely clear. Published single-particle cryo-EM structures of RTC appear to have resulted in various degrees of openness.

Here we were referring not to RTC structures in general, but to substrate-engaged RTCs in particular. The two substrate-engaged RTC structures under discussion in this paragraph are nearly identical (Figure 2c) despite large differences in substrate sequence (RhoTM2 vs preprolactin’s SP). We were surprised to find that this engaged structure creates noncovalent bonds between the Sec61 N-half and the ribosome. This bonding would tend to stabilise this particular engaged structure, and this stabilisation helps explain why the newly observed TMengaged structure is so similar to the previously observed SP-engaged structure. Without this stabilising N-half interaction, one might instead expect to see more variability, such as the reviewer suggests.

- A recent analysis of heimdallarchaea already hypothesized TRAP in these organisms and should be cited: Eme et al, Nature 618:992-999 (2023). The novel findings of this manuscript compared to Eme et al should be discussed.

We thank the reviewer for bringing this relevant contemporaneous work to our attention. Reviewing the putative TRAP homologs identified by Eme et al, we find that most do not in fact appear to be TRAP homologs at all, judged by the measures used in our work (reciprocal HHpred queries against the human proteome and predicted structural similarity). This is not surprising since Eme et al. relied on low-threshold sequence similarity searches rather than structural measures. To acknowledge this work, we have added a sentence as follows (italics): “To test whether these candidates are also similar to TRAPαβγ in sequence, we used them to perform reciprocal HHpred queries of the human proteome, and in each case the corresponding human TRAP protein was the top hit (E = 0.031 for TRAPα, 9.4×10-14 for TRAP β, and 110 for TRAPγ). A contemporaneous study has also claimed to find TRAP homologs in Heimdallarchaeota (Eme et al. 2023), although some caution is warranted in these assignments because they do not seem to share predicted structural similarity to TRAP subunits and do not find human homologs in reciprocal HHpred queries.”

- Given that the authors expand the evolutionary analysis of TRAP to archaea it would be helpful if sampling for RAMP4 were consistent (i.e., is TRAP present in the early eukaryotes that do not feature RAMP4? Is RAMP4 absent from heimdallarchaea?).

As stated in the text, RAMP4’s absence from early-branching eukaryotic taxa indicates that it was also absent from their archaeal ancestors. We did of course run such queries for completeness and indeed find no archaeal RAMP4. TRAP, for its part, is generally present in early-branching eukaryotic taxa, as stated in the text, and this necessarily includes those from which RAMP4 is absent.

- The authors may consider discussing (Gemmer et al 2023, https://doi.org/10.1101/2023.11.28.569136), which comes to similar conclusions for NEMO integration into the MPT.

We thank the reviewer for bringing this relevant work to our attention. We have added the following sentence to the section on NOMO: “Contemporaneous work has arrived at a similar model for PLD10-12 but did not model PLD1 (Gemmer et al. 2023).”

- The abundance approximation of RAMP4 in the native translocon by OccuPy should probably be taken with a grain of salt. The '80%' mentioned in the conclusion may stick around and could eventually turn out to be closer to 100%.

It is certainly possible that the occupancy of RAMP4 is higher than OccuPy estimates.

Unfortunately no available method can provide occupancy estimates with confidence intervals. The Western blots we have added to the revised manuscript are consistent with high occupancy, but cannot discriminate between 80 or 100%.

Minor- p. 5: The following sentence is incomplete: "Together, these factors explain why RAMP4's occupancy in prior cryo-EM maps was low enough to be overlooked, although in hindsight seems to be visible in several7,68,69"

Thank you for catching this typo. We have revised the sentence as follows: “Together, these factors explain why RAMP4's occupancy in prior cryo-EM maps was low enough to be overlooked, although in hindsight it is visible in several of them.”